# *SLC45A4* encodes a peroxisomal putrescine transporter that promotes GABA de novo synthesis

Cecilia Colson [1,4], Yujue Wang[1,2,3,4], James Atherton[1], Nisha Rani Dahiya[1], Davood Kharaghani [1] & Xiaoyang Su [1,2] ✉

Solute carriers (SLC) are membrane proteins that facilitate the transportation of ions and metabolites across either the plasma membrane or the membrane of intracellular organelles. With more than 450 human genes annotated as SLCs, many of them are still orphan transporters without known biochemical functions. We develop a metabolomic-transcriptomic association analysis, and we find that the expression of *SLC45A4* has a strong positive correlation with the cellular level of γ-aminobutyric acid (GABA). Using mass spectrometry and the stable isotope tracing approach, we demonstrate that SLC45A4 promotes GABA de novo synthesis through the Arginine/Ornithine/Putrescine (AOP) pathway. SLC45A4 functions as a putrescine transporter localized to the peroxisome membrane to facilitate GABA production. Taken together, our results reveal a biochemical mechanism where SLC45A4 controls GABA production.

Individual cells use channels and transporters embedded in the lipid membranes to exchange chemicals such as water, ions, nutrients, vitamins, cofactors, and many drugs with their environments[1,2]. Solute carrier (SLC) transporter family, comprised of 458 members distributed in 65 subfamilies[3], is the second-largest family of membrane proteins in the human genome, next to G-protein-coupled receptors (GPCRs)[4–8]. Not only are SLC transporters fundamental to cellular metabolism, but they also show clear links to human diseases. Based on Open PHACTS, a platform that integrates disease, chemical, and target databases, at least 190 different SLCs have been found mutated in human disease[9–11]. Despite increasing interest and the emergence of database ontology, around 30% of SLCs are still uncharacterized and remain orphan transporters with unidentified substrates, particularly when it concerns transporters with endogenous compounds[12].

In order to de-orphanize SLCs, we developed a metabolomic-transcriptomic association analysis, and we discovered that the expression of *SLC45A4* has a strong positive correlation with the cellular level of γ-aminobutyric acid (GABA). The function of SLC45A4 is largely unknown. It has been annotated as a proton(H+)-associated sucrose transporter based on its homology to the sucrose transporter

(SUC) genes in *Arabidopsis thaliana*[13]. In *Saccharomyces cerevisiae*, mouse *Slc45a4* restores the growth of a strain incapable of sucrose uptake. However, in a recent study, Meixner, E. et al. manually established a substrate-based ontology to identify potential substrates for orphan SLCs[14]. Their results suggest that SLC45A4 and its homolog SLC45A3 could be transporters for metal ions, but they did not identify carbohydrates as potential substrates. In mammalian cells, SLC45A4 was implicated in different types of cancers and associated with different clinical outcomes regarding the type of cancer. For example, *SLC45A4* gene expression is positively associated with tumor suppression in osteosarcoma[15] but with higher risks in cancers such as pancreatic ductal adenocarcinoma[16]. SLC45A4 is mostly described as a biomarker of the clinical severity of cancer, but its mechanisms are poorly understood.

GABA was first discovered in mammalian brains over half a century ago[17,18]. Besides its presence in the central nervous system, GABA has also been identified in many organs, including the pancreas, pituitary, testes, gastrointestinal tract, ovaries, placenta, uterus, and adrenal medulla[19]. The metabolic and signaling role of GABA in peripheral tissue is relatively poorly understood. GABA signaling has been

[1]Department of Medicine, Rutgers-Robert Wood Johnson Medical School, Rutgers University, New Brunswick, NJ, USA. [2]Rutgers Cancer Institute, New Brunswick, NJ, USA. [3]Present address: School of Pharmaceutical Sciences, Tsinghua-Peking Center for Life Sciences, Beijing Frontier Research Center for Biological Structure, Tsinghua University, Beijing, China. [4]These authors contributed equally: Cecilia Colson, Yujue Wang. ✉e-mail: xs137@rwjms.rutgers.edu

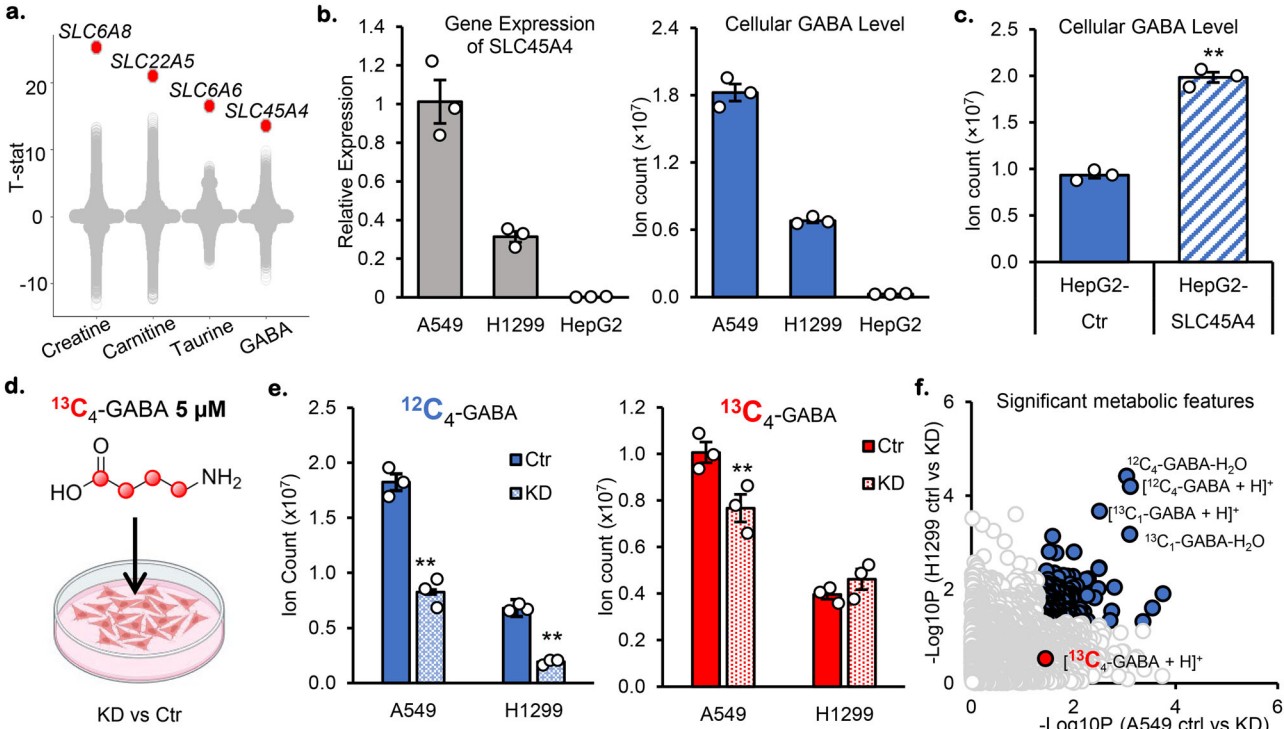

**Fig. 1 | SLC45A4 modulates cellular GABA levels but is not a GABA transporter.**
**a** Association between metabolites and the expression of 54271 genes. Each point represents a gene. Genes with the strongest association are colored in red.
**b** Relative mRNA expression of *SLC45A4* (left) and cellular GABA level (right) in A549, H1299 and HepG2 cells. **c** Cellular GABA level in HepG2 cells overexpressing empty vector (Ctr) or SLC45A4. **d** Schematic of $^{13}C_4$-GABA supplementation in cell media. **e** Cellular GABA level in A549 and H1299 cells transfected with scramble siRNA (Ctr) or si-SLC45A4 (KD) and cultured with 5 µM $^{13}C_4$ GABA for 24 h. Part of this figure was created in BioRender. Su, X. (2025) https://BioRender.com/0pfhaea.

**f** Metabolomes of A549 and H1299 cells transfected with scramble siRNA (Ctr) or si-SLC45A4 (KD) and cultured with 5 µM $^{13}C_4$ GABA for 24 h. Each point in the scatter plot represents a metabolite. Metabolites showing significant difference between Ctr and KD in both cell lines are colored in blue. In all figures, data are mean ± SEM; $n = 3$ biological independent replicates. **$P < 0.01$; *$P < 0.05$; ns, not significant by one-sided unpaired Student's t-test. The actual $p$-values are: **(c)**. $p = 0,00004$; **(e)**. for $^{12}C_4$-GABA, KD vs Ctr: A549 $p = 0,000381163$; H1299 $p = 0,000017$; for $^{13}C_4$-GABA, KD vs Ctr: $p = 0,015915824$. Source data are provided as Source Data file.

associated with stem cell proliferation[20], implicating its importance in cancer progression. Lastly, a study revealed that GABA treatment of mice developing triple-negative breast cancer promoted metastasis to the lung and brain[21]. There are two GABA synthesis pathways in mammals: in GABAergic neurons, GABA is synthesized by glutamic acid decarboxylase (GAD) enzymes[22]. In addition to this canonical GAD pathway, GABA can also be produced via the arginine-ornithine-putrescine (AOP) pathway[23,24]. In this work, we found, using stable isotope tracing, that the AOP pathway is the dominant pathway for GABA production in peripheral tissues, and SLC45A4 promotes GABA production through this pathway. Putrescine can be oxidized in tandem reactions catalyzed by diamine oxidases (DAOs)[25,26] or monoamine oxidases (MAOs)[27,28], and aldehyde dehydrogenases (ALDHs)[29,30] to produce GABA. Interestingly, dopaminergic neurons in the midbrain rely on the AOP pathway for GABA synthesis[31], demonstrating the importance of this pathway in GAD-negative cells and tissues.

Our findings reveal a new layer of the regulation of GABA metabolism, in which SLC45A4 works as a peroxisome putrescine transporter and promotes GABA de novo synthesis via the AOP pathway.

## Results

### Transcriptomic-Metabolomic association analysis reveals the role of SLC45A4 in GABA metabolism

Our overarching goal is to systematically investigate the biochemical functions of orphan SLC transporters. Our approach is to link the gene expression levels and the metabolite levels through a transcriptomic-metabolomic association analysis. We leveraged publicly available datasets from the Cancer Cell Line Encyclopedia

(CCLE)[32,33], and we built linear regression models to analyze the expression levels of 56,202 human genes and the cellular levels of 225 metabolites in 898 human cancer cell lines. For each model, we calculated the t-value for the slope. As expected, a number of SLC genes showed strong correlations with their cellular metabolite levels (Supplementary Fig. 1, Supplementary Data 1). For example, we found a strong positive correlation between cellular creatine levels and the expression levels of *SLC6A8* (Supplementary Fig. 2), which is a known creatine transporter[34]. Similarly, cellular carnitine levels are strongly correlated with the expression levels of *SLC22A5* (Fig. 1a), a known carnitine transporter[35]; and cellular taurine levels are strongly correlated with the expression levels of *SLC6A6* (Fig. 1a), a known taurine transporter[36]. Interestingly, we found that cellular GABA levels are strongly correlated with the expression levels of *SLC45A4* (Fig. 1a). *SLC45A4* shows a much stronger positive correlation with cellular GABA levels than the known GABA transporters (*GAT1:SLC6A1*, *GAT2:SLC6A13*, and *GAT3:SLC6A11*) and the GABA producing enzymes (*GAD1, GAD2*) (Supplementary Fig. 3). The current annotation of SLC45A4 as a proton(H$^+$)-associated sucrose transporter[13] is insufficient in explaining its role in GABA metabolism, so we set out to investigate the biochemical function of this gene.

To experimentally verify the transcriptomic-metabolomic associations identified from CCLE, we chose three human cell lines to work on: A549 and H1299 cells that express relatively high levels of *SLC45A4*, and HepG2 cells that express very low levels of *SLC45A4* as confirmed by relative mRNA quantification (Fig. 1b). We also measured the cellular GABA level using LC-MS. As predicted, we detected a lower level of GABA in HepG2 cells compared to the other two cell lines (Fig. 1b).

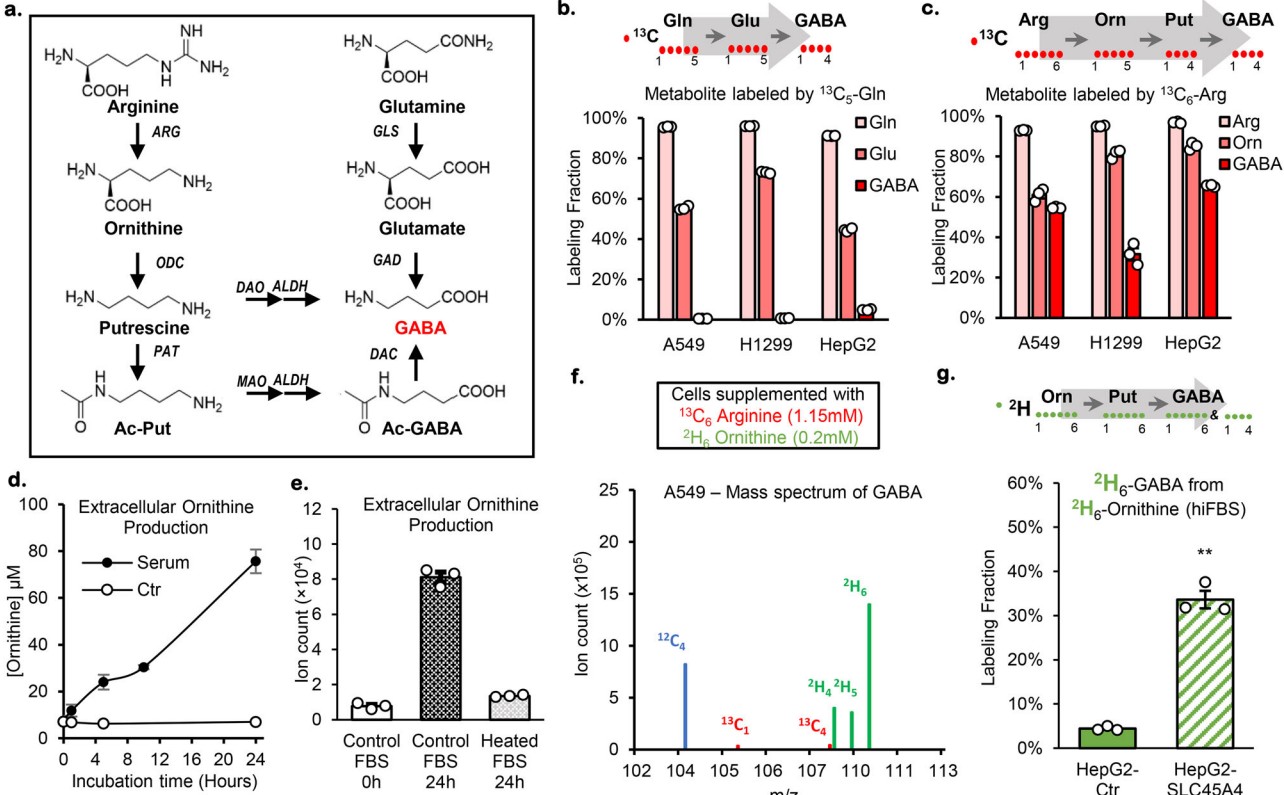

**Fig. 2 | Cellular GABA is produced from extracellular ornithine. a** GABA synthesis pathways. **b** $^{13}C_5$-Glutamine tracing experiment and cellular levels of $^{13}C_5$-Glutamine, $^{13}C_5$-Glutamate and $^{13}C_4$-GABA. **c** $^{13}C_6$-Arginine tracing experiment and cellular levels of $^{13}C_6$-Arginine, $^{13}C_5$-Ornithine and $^{13}C_4$-GABA after $^{13}C_6$-Arginine tracing experiment. **d** Relative quantification of extracellular ornithine production in fresh RPMI media (Ctr, empty circles) or containing 10% fetal bovine serum (Serum, full circles). **e** Ornithine levels after 24 h in RPMI media containing 10% of control fetal bovine serum (Control FBS) or after heat-inactivation (Heated FBS).

**f** Simultaneous labeling of GABA from 1.15 mM $^{13}C_6$-arginine and 0.2 mM $^2H_6$-ornithine in A549 cells cultured heat-inactivated dialyzed FBS (hiFBS). **g** Cellular $^2H_6$-GABA levels in HepG2 cells overexpressing empty vector (Ctr) or SLC45A4 and cultured with 10% hiFBS and 0.2 mM $^2H_6$-ornithine. In all figures, data are mean ± SEM; $n = 3$ biological independent replicates. **$P < 0.01$; *$P < 0.05$; ns, not significant by one-sided unpaired Student's t-test. The p-value in (**g**). is $p = 0.00006$. Source data are provided as Source Data file.

The absolute quantitation using $^{13}C$ internal standard showed that the GABA concentration in A549 cells is 1.67 nmols/$10^6$ cells (Supplementary Fig. 4a). To demonstrate that the expression level of *SLC45A4* could directly affect cellular GABA level, we transiently overexpressed (OE) *SLC45A4* in HepG2 cells. The LC-MS results showed a higher level of GABA in the *SLC45A4* OE cells (Fig. 1c), demonstrating that SLC45A4 controls cellular GABA levels.

## SLC45A4 increases cellular GABA levels by facilitating GABA de novo synthesis

Next, we inquired: how does SLC45A4 affect cellular GABA levels? First, we tested the hypothesis that SLC45A4 works as a sucrose transporter. A549 and H1299 cells were cultured in $^{13}C_6$-glucose with or without supplemented $^{12}C$-sucrose. If sucrose is catabolized by these cells, it should be incorporated into the glycolytic and the TCA cycle intermediates, and therefore decrease the $^{13}C$ enrichment in these metabolites. Our observations did not support the sucrose-transporting activity. While we observed a robust $^{13}C$ labeling in glycolysis and TCA cycle intermediates, the addition of sucrose had no negative impact on them, suggesting sucrose is not catabolized in A549 and H1299 cells with the basal expression of SLC45A4 (Supplementary Fig. 5). While our experiment does not completely rule out the sucrose-transporting activity of SLC45A4 in the absence of glucose, the regulation of GABA synthesis is likely not due to the sucrose-transporting activity.

The next logical hypothesis is that SLC45A4 functions as a GABA transporter facilitating GABA uptake. To evaluate this potential GABA transporter activity, we fed stable isotope-labeled GABA ($^{13}C_4$-GABA) to

the control and the *SLC45A4* Knock Down (KD) A549 and H1299 cells (Fig. 1d, e). In both cell lines, we observed decreases in total GABA levels when *SLC45A4* was knocked down, confirming the role of SLC45A4 in GABA metabolism. However, we found that the decrease in GABA level was mainly due to a decline in endogenous, unlabeled GABA ($^{12}C_4$-GABA) rather than a decline in the uptake of $^{13}C_4$-GABA (Fig. 1e). Our untargeted metabolomics analysis picked up 2,676 features from this dataset (Fig. 1f). The top 4 hits that are most significantly changed in the KD cells are all related to unlabeled GABA (the protonated ion $[M + H]^+$, the protonated ion with a water loss $[M + H-H_2O]^+$, $^{13}C$ isotope natural abundance ion $[^{13}C_1-M + H]^+$, and $^{13}C$ isotope natural abundance ion with a water loss $[^{13}C_1-M + H \cdot H_2O]^+$). The $^{13}C_4$-GABA, which indicates GABA uptake, was significantly but slightly changed in A549 KD cells but not in H1299 KD cells. These results indicate that although it regulates cellular GABA levels, SLC45A4 does not function directly as a GABA transporter. From these results, we inferred that SLC45A4 is required for the de novo synthesis of GABA.

## Ornithine is the major source of GABA de novo synthesis in human cancer cells

To investigate the role of SLC45A4 in GABA de novo synthesis, we explored different GABA biosynthetic routes in A549 and H1299 cells. The most well-studied GABA synthesis route is through glutamate decarboxylation (GAD pathway) in GABAergic neurons (Fig. 2a), while other tissues have been reported to use arginine, ornithine, or putrescine as sources for GABA (AOP pathway)[37]. Therefore, we fed the cells with either $^{13}C_5$-glutamine or $^{13}C_6$-arginine as the substrate for

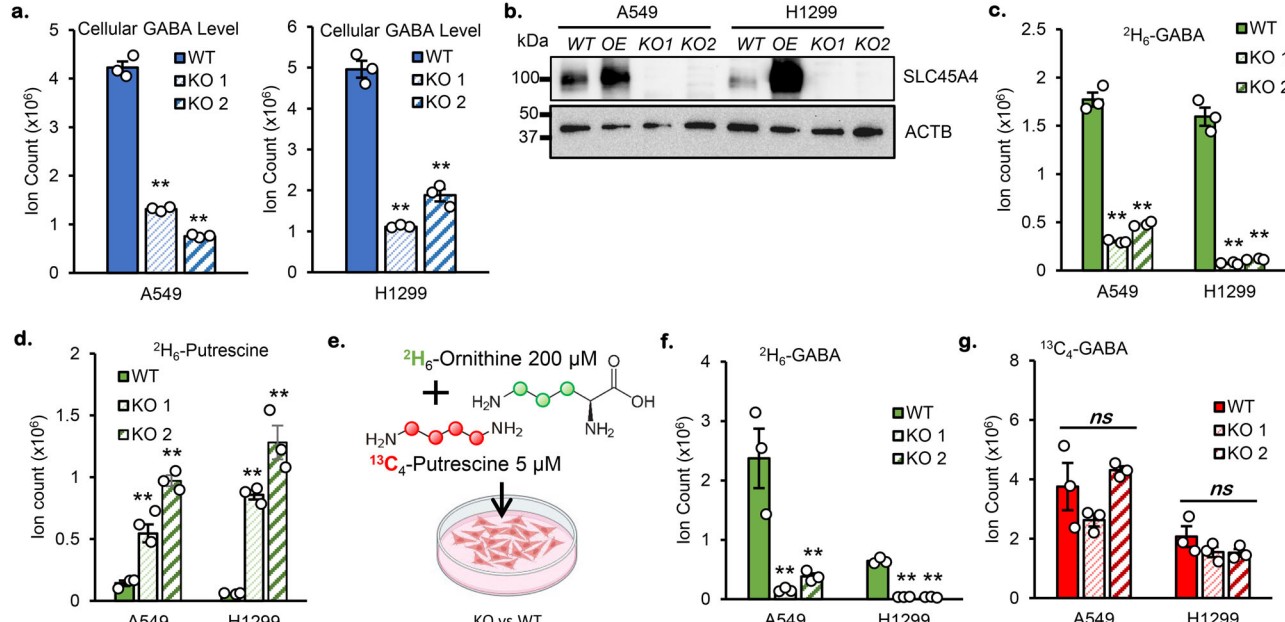

**Fig. 3 | GABA production from ornithine and putrescine is differentially regulated by SLC45A4. a** Cellular level of GABA in A549 and H1299 Wild-type (WT) and their 2 homozygotes generated SLC45A4-knockouts (KO1 & KO2). **b** SLC45A4 protein expression in A549 and H1299 WT, overexpressing SLC45A4 (OE) and KOs cells. **c, d** Cellular level of GABA (**c**) and Putrescine (**d**) derived from $^2H_6$-ornithine in A549 and H1299 WT and KO cells. **e** Schematic of $^2H_6$-ornithine and $^{13}C_4$-putrescine dual-tracing experiment. Part of this figure was created in BioRender. Su, X. (2025) https://BioRender.com/0pfhaea. **f, g** Cellular level of GABA derived from (**f**). $^2H_6$-ornithine ($^2H_6$-GABA) and from (**g**). $^{13}C_4$-putrescine ($^{13}C_4$-GABA) in dual-tracing experiment. In all figures, data are mean ± SEM.; $n = 3$ biological independent

replicates. **$P < 0.01$; *$P < 0.05$; ns, not significant by one-sided unpaired Student's t-test (**a, c, d**) or one-way ANOVA followed by Tukey's post-hoc test for dual-tracing experiment (**f, g**). The actual $p$-values are: (**a**) KOs vs WT: A549 $p_{KO 1} = 0,0025$; $p_{KO 2} = 0,00085$; H1299 $p_{KO 1} = 0,0000002$; $p_{KO 2} = 0,0000018$ (**c**). KOs vs WT: A549 $p_{KO 1} = 0,00001$; $p_{KO 2} = 0,00003$; H1299 $p_{KO 1} = 0,00004$; $p_{KO 2} = 0,00004$; (**d**) KOs vs WT: A549 $p_{KO 1} = 0,001$; $p_{KO 2} = 0,00003$; H1299 $p_{KO 1} = 0,00001$; $p_{KO 2} = 0,0004$. **f** A549: F(2) = 55,08 $p$ = 3.05e-11; H1299: F(2) = 37,73 $p$ = 2.98e-09 (**g**). A549: F(2) = 0,22 $p$ = 0,803; H1299: F(2) = 0,176 $p$ = 0,839. Source data are provided as Source Data file.

GABA synthesis, and we measured the $^{13}C$ incorporation in GABA (Fig. 2b, c). While supplemented $^{13}C_5$-glutamine produced a significant amount of $^{13}C_5$-glutamate (50–70%), it labeled less than 1% of GABA in A549 and H1299 cells (Fig. 2b), suggesting that glutamate decarboxylases (GADs) do not contribute significantly to GABA synthesis in these cells. On the other hand, $^{13}C_6$-arginine labeled >30% GABA with $^{13}C$ (Fig. 2c), suggesting that the AOP pathway is the dominant pathway for GABA synthesis in A549 and H1299 cells. In fact, our transcriptomic-metabolomics association study not only identified *SLC45A4* as the top gene related to GABA production, but it also revealed that ornithine decarboxylase (*ODC1*), which converts ornithine to putrescine, is the gene with the second-highest positive correlation with cellular GABA levels out of >56,000 transcripts assessed (Supplementary Fig. 3c). These results support the idea that SLC45A4 promotes GABA synthesis through the AOP pathway.

Interestingly, when culturing cells in $^{13}C_6$-arginine, we observed a robust production of $^{13}C_5$-ornithine (Fig. 2c), which raised the question of whether ornithine is an important substrate for GABA de novo synthesis in the presence of arginine. Although ornithine is not proteogenic and is often omitted from most cell culture media, it was found to be present at roughly 200 μM in plasma and 300 μM in the tumor interstitial fluid[38]. In cell culture, the extracellular production of ornithine happened because the fetal bovine serum (FBS) we used contains arginase activity, which breaks down the arginine in the RPMI 1640 media to ornithine (Fig. 2d). To avoid the extracellular ornithine production, we heat-inactivated the FBS at 70 °C, which is effective in eliminating arginase activity (Fig. 2e). Using the heat-inactivated FBS (hiFBS), we prepared RPMI media containing $^{13}C_6$-arginine (1.15 mM) and $^2H_6$-ornithine (200 μM). Our results showed that in A549 and H1299 cells, 98% of labeled GABA were $^2H$-labeled (both $^2H_6$- and $^2H_4$-GABA were generated due to the molecular symmetry of putrescine)

(Fig. 2f and Supplementary Fig. 6) which suggested that they were derived from ornithine. Less than 2% of labeled GABA were $^{13}C$-labeled indicating that only a small fraction of GABA was derived from arginine. These observations indicated that ornithine is the preferred substrate for GABA biosynthesis.

## SLC45A4 promotes GABA de novo synthesis from ornithine and putrescine

Knowing that GABA de novo synthesis mostly comes from ornithine, we first tested whether the overexpression of SLC45A4 would enhance GABA production from ornithine. Indeed, when *SLC45A4* was overexpressed in HepG2 cells, a greater amount of $^2H_6$-GABA made from $^2H_6$-ornithine was detected (Fig. 2g). The next question is whether SLC45A4 is strictly required for GABA synthesis from ornithine. To answer this question, we generated *SLC45A4* KO cell lines in both A549 and H1299 backgrounds (Fig. 3a, b). Then, we cultured the WT and KO cells with $^2H_6$-ornithine, and we observed that the *SLC45A4* KO significantly decreased the production of GABA from ornithine, but did not completely eliminate it (Fig. 3c). Interestingly, we also observed the accumulation of $^2H_6$-putrescine in the KO cells (Fig. 3d), suggesting putrescine oxidation might be impeded in *SLC45A4* KO cells.

To explore the role of SLC45A4 in ornithine and putrescine metabolism, we performed a $^2H_6$-ornithine and $^{13}C_4$-putrescine dual-tracing experiment (Fig. 3e). Putrescine, the decarboxylation product of ornithine, is the downstream intermediate for GABA biosynthesis. Interestingly, we observed a differential effect of *SLC45A4* KO on ornithine and putrescine metabolism in GABA production. As shown in Fig. 3f, *SLC45A4* KO significantly decreased the level of $^2H_6$-GABA. However, *SLC45A4* KO did not affect the level of $^{13}C_4$-GABA (Fig. 3g), which suggests that SLC45A4 may not be required for extracellular putrescine oxidation to produce GABA. These results create a

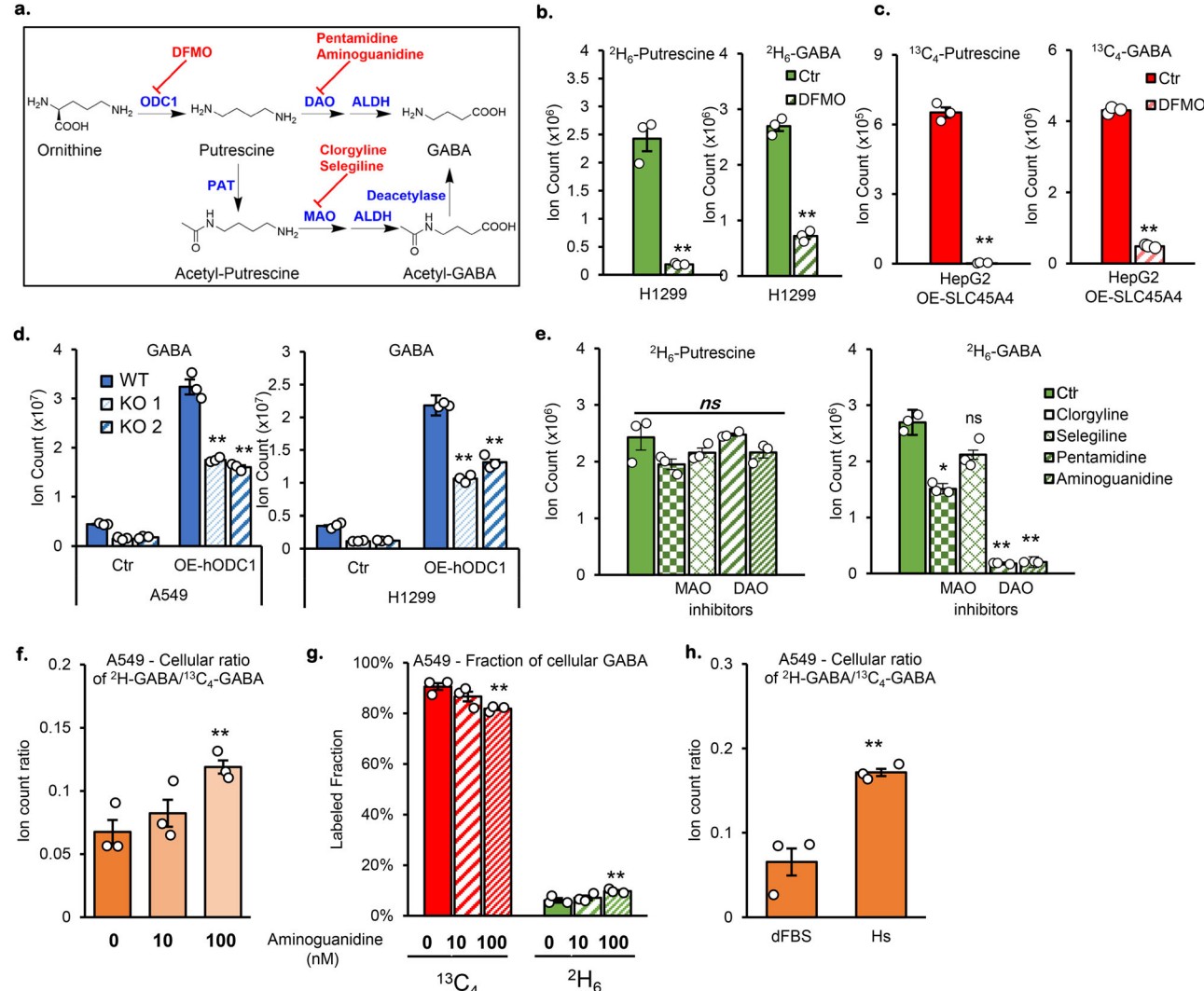

**Fig. 4 | GABA synthesis depends on the activity of ornithine decarboxylase and diamine oxidases. a** Inhibition of GABA synthesis by DFMO or Monoamine Oxidases/Diamine Oxidases (MAO/DAO) inhibitors. **b, c** Cellular level of Putrescine (left) and GABA (right) after 24 h treatment with ODC1-inhibitor DFMO (100 μM) in (**b**) H1299 culture with $^2H_6$-Ornithine and (**c**) HepG2 overexpressing SLC45A4 cultured with $^{13}C_6$-Arginine. **d** Cellular level of GABA in A549 (left) and H1299 (right) WT and their SLC45A4-KO cells (KO1 & KO2) overexpressing empty vector (Ctr) or human-ODC1 (OE-hODC1). **e** Cellular level of putrescine (left, $^2H_6$-putrescine) and GABA (right, $^2H_6$-GABA) in H1299 cells cultured with $^2H_6$-Ornithine and treated with 0 μM (Ctr) or MAO inhibitors Clorgyline (5 μM) or Selegiline (10 μM) or DAO inhibitors Pentamidine (1 μM) or Aminoguanidine (0.1 μM). **f** Cellular ratio of $^2H$-GABA/$^{13}C_4$-GABA in A549 WT cells supplemented with $^2H_6$-Ornithine and $^{13}C_4$-Putrescine in absence (0) or presence of 10 or 100 nM of aminoguanidine.

**g** Fractions of cellular GABA in A549 WT cells supplemented with $^2H_6$-Ornithine and $^{13}C_4$-Putrescine in absence (0) or presence of 10 or 100 nM of aminoguanidine. **h** Cellular ratio of $^2H$-GABA/$^{13}C_4$-GABA in A549 cells supplemented with $^2H_6$-Ornithine and $^{13}C_4$-Putrescine in regular serum (dFBS) or horse serum (Hs). In all figures, data are mean ± SEM.; $n = 3$ biological independent replicates. **$P < 0.01$; *$P < 0.05$; ns, not significant by two-sided (**b–e**) or one-sided (**f–h**) unpaired Student's t-test. The actual p-values are: (**b**) Ctr vs DFMO $p_{2H6\text{-}put} = 0,0005$; $p_{2H6\text{-}GABA} = 0,00004$; (**c**) Ctr vs DFMO $p_{13C4\text{-}put} = 0,000007$; $p_{13C4\text{-}GABA} = 0,0000003$; (**d**) Ctr vs OE: A549 $p_{WT} = 0,00005$; $p_{KO1} = 0,000001$; $p_{KO2} = 0,000004$; H1299 $p_{WT} = 0,0000005$; $p_{KO1} = 0,000009$; $p_{KO2} = 0,00003$; (**e**) $^2H_6$-GABA, Ctr vs inhibitors: $p_{clor} = 0,0003$; $p_{penta} = 0,00001$; $p_{amino} = 0,00001$; (**f**) $p = 0,008$; (**g**) $p_{13C4} = 0,003$; $p_{2H6} = 0,01$; (**h**) $p = 0,003$. Source data are provided as Source Data file.

conundrum. On one hand, *SLC45A4* KO results in an intracellular build-up of putrescine and decreased GABA production, pointing to a bottleneck in putrescine oxidation. On the other hand, *SLC45A4* KO cells show no deficiency in converting supplemented putrescine to GABA. To resolve this conundrum, we explored another approach to modulate intracellular putrescine. As mentioned before, *ODC1* has the second-highest correlation with GABA levels, as revealed by our transcriptomic-metabolomic analysis. In fact, inhibition of ODC1 by difluoromethylornithine (DFMO) (Fig. 4a) decreases cellular putrescine and GABA levels in both H1299 cells and in HepG2 cells overexpressing *SLC45A4* (Fig. 4b, c). In parallel, *ODC1* overexpression increases both cellular putrescine and GABA levels (Fig. 4d). However, the *ODC1* overexpression did not fully rescue the deficiency in GABA

production in *SLC45A4* KO cells, suggesting SLC45A4 is required for efficient putrescine oxidation. It is also noteworthy that *ODC1* overexpression increased the extracellular putrescine concentration in both WT and SLC45A4 KO cells (Supplementary Fig. 7b). Therefore, the increase of GABA in *SLC45A4* KO after *ODC1* overexpression could come from the oxidation of extracellular putrescine, which does not depend on SLC45A4. Overall, the results from putrescine supplementation and *ODC1* overexpression suggest SLC45A4 is required for cellular putrescine oxidation, and it hinted to us to consider how the intracellular and extracellular putrescine is differentially oxidized.

Depending on the cell types, GABA can be produced either through DAO or MAO. To test which pathway is active in A549 and H1299 cells, we treated the cells with various DAO and MAO inhibitors

(Fig. 4a). Pentamidine and aminoguanidine are inhibitors of diamine oxidases[39,40], which are Cu(II)-dependent enzymes. Clorgyline is a specific inhibitor of MAO-A[41], and selegiline is a specific inhibitor of MAO-B[42], both are flavin adenine dinucleotide (FAD)-dependent enzymes[43]. While clorgyline induced a small decrease in $^2$H-GABA, selegiline showed no effect on GABA production when both the DAO inhibitors more profoundly blunted GABA production from $^2$H$_6$-ornithine (Fig. 4e). These results suggest GABA is produced from putrescine through DAO activities. One potential confounding factor is the diamine oxidase activity in FBS. Although some reports argued that the purified serum DAO shows poor activity on putrescine[44,45], the putrescine oxidation activity was found in FBS[46]. Therefore, we suspect that this enzymatic activity contributes to the oxidation of extracellular putrescine nonetheless. To study the cellular GABA production pathway without the interference of extracellular diamine oxidase activity, we explored two options. First, we supplemented the cells with $^2$H$_6$-ornithine and $^{13}$C$_4$-putrescine treated with 10 nM or 100 nM or without the DAO inhibitor, aminoguanidine. As previously shown in Fig. 4e, aminoguanidine decreases total cellular GABA but $^2$H/$^{13}$C ratio increases by 2 folds (Fig. 4f, Supplementary Fig. 8a), indicating that GABA derived from extracellular putrescine ($^{13}$C fraction) is significantly reduced (10% reduction, Fig. 4g) while GABA derived from intracellular putrescine ($^2$H fraction) is significantly augmented (2 folds, Fig. 4g and Supplementary Fig. 8a). Second, we cultured the cells under regular dialyzed heat inactivated FBS (dFBS) or Horse serum (Hs), which has been shown to have a much lower DAO activity[47]. Our results show that, similar to aminoguanidine, $^2$H/$^{13}$C ratio of cellular GABA is strongly increased, from 3 to 15 folds (Fig. 4h and Supplementary Fig. 8b). These results suggest that supplemented putrescine is oxidized extracellularly and that the production of $^2$H-GABA depends, at least partially, on the intracellular DAOs. We could speculate that this will explain why supplemented putrescine is converted into GABA in *SLC45A4* KO cells whereas putrescine derived from ornithine is not. This suggests that SLC45A4 promotes GABA from ornithine and intracellular putrescine, while extracellular putrescine is oxidized in a SLC45A4-independent manner.

### *SLC45A4* encodes a peroxisomal putrescine transporter

We next wondered about SLC45A4's location since a number of SLC transporters are known to be intracellular transporters. For example, the members of the SLC25 family are mostly mitochondrial transporters[48], and the members of the SLC35 family are mostly localized to Golgi and endoplasmic reticulum (ER)[49]. SLC45A2, another member of the SLC45 family, has been found to be a melanosome transporter[50,51]. We used immunofluorescence microscopy to investigate the subcellular localization of SLC45A4. First, we tested the specificity of a commercial SLC45A4-targeting antibody by transiently overexpressing *SLC45A4* with an HA-tag and observed perfect colocalization between green signals (anti-HA) and red signals (anti-SLC45A4) (Supplementary Fig. 9). Therefore, we used this antibody to detect endogenous SLC45A4 transporter in A549 cells. SLC45A4 (green signal) is clearly detected around the cell nucleus, without overlapping actin signal (Fig. 5a, top panel), suggesting organellar localization rather than plasma membrane localization. This is also confirmed by the absence of co-localization with another plasma membrane marker, E-cadherin, as shown in Supplementary Fig. 10a. Moreover, SLC45A4 signal strongly co-localized with ABCD3, which is a peroxisomal transporter of long-chain fatty acids and is frequently used as a peroxisomal marker (Fig. 5a, bottom panel). In addition, we observed no overlap between SLC45A4 signals and Golgi marker, Giantin; endoplasmic reticulum marker, Calnexin or mitochondrial marker, Mitotracker (Supplementary Fig. 10b).

To further validate the subcellular localization of SLC45A4, we isolated peroxisomes in A549 cells by using two different methods, either with an affinity tag (Fig. 5b–d) or with the more traditional

density gradient centrifugations (Supplementary Fig. 10a, b). We first used A549 cells stably expressing a truncated peroxisome membrane protein PEX26 tagged with e-GFP and 3xHA[52] (Fig. 5b) confirmed that the chimeric protein co-localizes with a specific peroxisome membrane-bound protein ABCD3 (Fig. 5c). A549 cells PEX26-eGFP-3xHA were submitted to hypotonic lysis and immunoprecipitation with magnetic beads tagged with anti-HA antibody, following the protocol developed in Sabatini's lab[52] (Fig. 5d, top schematic) and the results showed a strong enrichment of SLC45 A4 protein in the HA-IP fraction, associated with an enrichment of peroxisome-specific proteins ABCD3 and catalase. These results were corroborated with gradient density centrifugation protocol (Supplementary Fig. 10a) which showed similar SLC45A4 enrichment in the fractions containing peroxisome specific proteins, ABCD3 and catalase (Supplementary Fig. 10b; lanes L1 and L2) compared to lysates pre-gradient separation (Supplementary Fig. 10b; lanes WC, P1 and Sup1).

Finally, we confirmed the putrescine-transporting activity of SLC45A4. We used a wheat-germ cell-free translation system to make reconstituted proteoliposomes containing SLC45A4 or a chimeric membrane protein eGFP-OMP25 (Fig. 5e). We performed a $^3$H-putrescine uptake assay and our results show that SLC45A4-containing proteoliposome has significantly higher putrescine uptake activity compared to the control proteoliposomes (Fig. 5e).

Taken together, our results revealed that SLC45A4 functions as a peroxisomal putrescine transporter to support GABA synthesis.

## Discussion

CCLE Database is an invaluable resource for the multi-omics characterization of human cancer cell lines[32,33,53]. Previous works have identified point mutations, copy number variations, and DNA methylations that may impact cellular metabolite levels[33]. Interestingly, a recent study showed that the cellular level of 1-methylnicotinamide (1-MNA) is strongly correlated with nicotinamide N-methyltransferase (NNMT) expression, its synthesizing enzyme[54]. In our study, we developed a transcriptomic-metabolomic association analysis that exposed several transporter genes strongly correlating with the level of their cellular metabolites and revealed the function of SLC45A4 in GABA metabolism. These discoveries demonstrate the utility of the CCLE database in understanding fundamental biochemistry.

Our transcriptomic-metabolomic association analysis established that cellular levels of creatine, carnitine, and taurine are mostly determined by the expression of their corresponding transporters. This result indicates that in many cases, in vitro cultured cells don't perform de novo synthesis of these metabolites but instead take them from the media, presumably from the serum component. In light of this, it is conceivable that dialyzed FBS, compared to regular FBS, may affect cell metabolism due to its lack of key metabolic factors. Meanwhile, one potential caveat of our association analysis is that it is based on the metabolomic profiles of cancer cells cultured under common media such as DMEM or RPMI 1640. It would be important to validate such correlations using either primary cells and/or culture media resembling tumor interstitial fluid.

Despite the fact that SLC45A4 has a high correlation with the cellular GABA level, unlike other transporters showing high correlations with their cellular substrates, SLC45A4 is not a GABA transporter. This observation shows that the substrate selectivity and the metabolic consequence of a transporter can be very different. This seems to be true, particularly for subcellular transporters. MFSD12, for example, mediates the import of cysteine into lysosomes and melanosomes and is required for the synthesis of the red skin pigment pheomelanin[55]. SLC22A14, a mitochondrial riboflavin transporter, is required for fatty acid oxidation and ATP production, and would eventually affect spermatozoa function and male fertility. Therefore, it is important to understand both the biochemical activities of SLC transporters and their metabolic consequences.

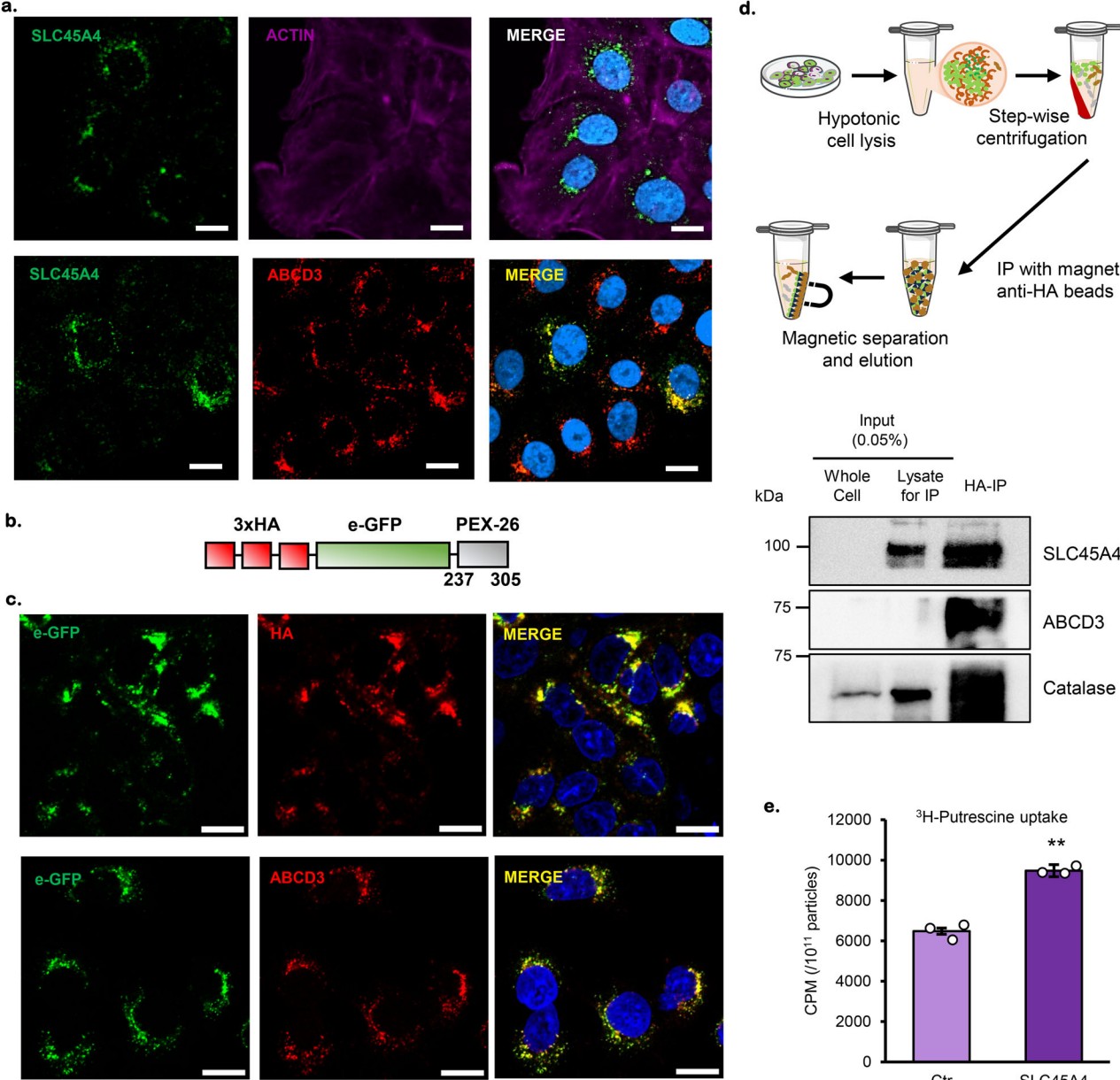

**Fig. 5 | SLC45A4 encodes a peroxisomal putrescine transporter.**
**a** Immunofluorescent detection of SLC45A4 subcellular localization: SLC45A4 endogenous, green; Actin, cytoskeleton marker, magenta; ABCD3, peroxisome marker, red; nucleus marker, DAPI, blue. **b** Schematic of the construct containing a truncated membrane-bound peroxisome protein PEX26 (237-305) tagged with e-GFP and 3-HA. **c** A549 cells stably expressing the PEX26-eGFP-3xHA (top panel) co-localizing with peroxisome marker ABCD3 (bottom panel). **d** Simplified workflow for HA immunoprecipitation (HA-IP) from A549 cells stably expressing PEX26-eGFP-3xHA and SDS-PAGE of whole cell, total lysate and HA-IP fraction: SLCC45A4

was detected with markers of peroxisome, ABCD3 and catalase. Image adapted from Servier Medical Art (https://smart.servier.com/), licensed under CC BY 4.0 (https://creativecommons.org/licenses/by/4.0/). **e** Putrescine uptake in SLC45A4-containing proteoliposomes (SLC45A4) and Control (Ctr, OMP25-containing proteoliposomes) in presence of $^3$H-putrescine. In all figures, data are mean ± SEM; $n = 3$ biological independent replicates. **$P < 0.01$ by one-sided unpaired Student's t-test. Micrographs in (**a**, **c**). are representatives' images from 3 independent experiments. Scale bar 10 μm. The actual $p$-values is (**e**). Ctr vs SLC45A4 $p = 0.0001$. Source data are provided as Source Data file.

Literature on the SLC45A4 has been relatively scarce and mainly focuses on gene expression and associated phenotype, as we are the first to investigate its biochemical function. So far, *SLC45A4* expression has been linked to clinical outcomes in various cancers[15,16] but remained an orphan transporter. In this study, we discovered that SLC45A4 promotes GABA de novo synthesis in human cancer cells. Whether these effects depend on the GABA signaling awaits further studies. Meanwhile, SLC45A4 likely plays an important role in regulating GABA synthesis in all GAD-negative cells. The role of SLC45A4 in glial cells[56] and non-GABAergic neurons awaits further study.

One limitation of our study is that we don't have a comprehensive analysis of cellular polyamines, including spermidine and spermine. This is due to the fact that these compounds are too polar to be eluted well from HILIC columns. These compounds are typically chemically derivatized to be more accurately quantified on LC-MS. Given our results that SLC45A4 is a peroxisome putrescine transporter, it is possible that the polyamine synthesis pathway is also under the regulation of SLC45A4. In fact, the polyamine spermidine (downstream of putrescine) is directly involved in proliferation by its role in the hypusination of *EIF5A* (Eukaryotic Translation Initiation Factor 5A)[57,58]. Further studies on the role of SLC45A4 on polyamine metabolism are warranted.

Finally, the function of SLC45A4 may exceed the transporter protein it encodes. Through alternative splicing, the genetic locus of *SLC45A4* produces a circular RNA named *circSLC45A4*. *circSLC45A4*, which is unlikely to encode a fully functional transporter, is highly expressed in the human fetal cortex and is required to keep neural cells in a progenitor state[59]. The physiological role of SLC45A4 is multifaceted.

Taken together, our results demonstrate that the transcriptomic-metabolomic association analysis is capable of revealing genes responsible for cellular metabolite accumulation, and the identification of SLC45A4 as a new regulator of GABA de novo synthesis exemplifies its utility.

## Methods

### Metabolomic-Transcriptomic association calculation
The metabolomics and transcriptomic profiles of 898 cancer cell lines were downloaded from the Cancer Cell Line Encyclopaedia (CCLE)[60] portal (https://portals.broadinstitute.org/ccle). For each pair of the gene and the metabolite, a linear regression model was applied to evaluate the association in the 898 cell lines. The T statistics (t-stat) and P values were calculated.

### Cell culturing
A549, H1299, and HepG2 cells were purchased from American Type Culture Collection (ATCC). The HEK293T cells were a gift from Dr. Wei-Xing Zong of Rutgers University. A549 and H1299 cell lines were cultured in RPMI 1640 media (Gibco, 11875135), HepG2 and HEK293T in DMEM media (ATCC, 30-2003); all media are supplemented with 10% FBS (Gibco, 26140079) and 1% Penicillin-Streptomycin (Gibco, 15140122). When appropriate, the FBS was substituted with dialyzed FBS (Gibco, A3382001) heat-inactivated at 70 °C for 1 h or with Horse serum (Gibco). For isotope labeling experiments that required substituting arginine or glutamine with U-$^{13}$C isotopologues, the corresponding substances were supplemented to SILAC RPMI 1640 Flex Media (Gibco, A2494201) to match the same formula as ordinary RPMI 1640 (*See Stable Isotope Tracing section*). For DAO and MAO inhibition, the following inhibitors were used: Aminoguanidine (Neta Scientific, AST-E79081), Pentamidine (Sigma, P0547), Clorgyline (Cayman,15925), Selegiline (Spectrum, S15981GM) and DFMO (Cayman, 16889). All cell lines were grown in standard cell culture conditions (37 °C, 5% $CO_2$) and were free of microbial contamination, including mycoplasma. Cells were passaged twice a week and used between passages 10 to 20.

### Expression plasmids
Human cDNA encoding SLC45A4 (NCBI: NM_001286646.1) was purchased from GenScript. The DNA sequence of SLC45A4 was cloned to pLPC-N FLAG vector (Addgene, 12521) using BamHI-HF (NEB, R3136S) and EcoRI-HF (NEB, R3101S) restriction digestion and Quick Ligation Kit (NEB, M2200S). Sequences coding 3×HA or 3×HA-GFP tags were added to the C-terminal of the protein using gene-specific PCR and HiFi assembly (NEB). The 3×HA and 3×HA-GFP sequences were cloned from pMXs-3XHA-EGFP-OMP25 plasmid (Addgene 83356).

pLJC5-3XHA-EGFP-PEX26 was a gift from David Sabatini (Addgene, 139054; http://n2t.net/addgene:139054). The DNA sequence of 3XHA-EGFP-PEX26 was cloned to pLPC-N FLAG vector (Addgene, 12521) using BamHI-HF (NEB, R3136S) restriction digestion, gene-specific PCR and Quick Ligation Kit (NEB, M2200S).

For the proteoliposome, DNA sequences of SLC45A4 or EGFP-OMP25 (derived from Addgene, 83356) were cloned to pEU-E01 vector (CellFree Sciences) using EcoRV-HF (NEB, R3136S) and Q5® Hot Start High-Fidelity DNA Polymerase (NEB, M0493) with appropriate primers. The 3x-HA tag was added to the C-terminus of the expressed protein.

Human cDNA encoding ODC1 (NCBI: NM_002539.3) was purchased from GenScript with eGFP cloned at the C-terminal in the vector pcDNA3.1.

### Transient Overexpression
Cells were seeded in new 12-well plates at the density of $2 \times 10^5$/ml. When the cells reached about 70% confluence, cells were transfected with Lipofectamine 3000 according to manufacturer's instructions (Invitrogen, L3000015). 1 µg expression vector plasmid DNA was first mixed with 2 µl P3000 reagent in 50 µl Opti-MEM reduced serum medium (Gibco, 31985070), then mixed with 3 µl Lipofectamine 3000 in another 50 µl Opti-MEM. The mixture was incubated at room temperature for 15 min and added directly to the cell culture. 24 h later, the media were replaced with the one containing stable isotope tracers if necessary.

### Retroviral production and transduction
HEK293T cells were seeded in new T-75 flask at the density of $2 \times 10^5$/ml. When the cells reached about 70% confluence, cells were transfected with Lipofectamine 3000 (Invitrogen, L3000015) according to manufacturer's instructions. 15 µg expression vector plasmid, 9 µg pUMVC plasmid (Addgene, 8449), and 3 µg pCMV-VSV-G plasmid (Addgene, 8454) was used for each transfection. The plasmid DNA was first mixed with 40 µl P3000 in 750 µl Opti-MEM reduced serum medium (Gibco, 31985070), then mixed with 59 µl Lipofectamine 3000 in another 750 µl Opti-MEM. The mixture was incubated at room temperature for 15 min and added directly to the cell culture. The culture media were refreshed with normal DMEM with 10% FBS and 1x Penicillin-Streptomycin after 6 h. 24 h and 48 h after transfection, media were collected, centrifuged for 5 min at 1000 × g and filtered using a 0.45 µm filter. The titer of virus was measured using the qPCR Retrovirus Titer Kit (ABM, G949). If the titer was less than $2 \times 10^6$ IU/ml, the virus media was concentrated using the Retro-X Concentrator (Takara, 631455) to reach $2 \times 10^6$ IU/ml. The virus media was aliquoted and stored at −80 °C. Cells to be transduced were seeded at the density of $2 \times 10^5$/ ml. When the cells reached about 50% confluence, conditioned media with virus was mixed with normal cell media with Multiplicity Of Infection (MOI) 1:1 ratio and supplemented with 10 ug/ml polybrene infection transfection reagent (Sigma, TR-1003). The mixture with appropriate volume was added to the cell three times with 12-18 h interval. After 7 days, cells were sorted for GFP$^+$ if necessary.

### siRNA knockdown
A549 or H1299 cells were seeded in 6-well plates at the density of $2 \times 10^5$/ ml. When the cells reached about 70% confluence, cells were transfected with siRNA Transfection Reagent (Santa Cruz Biotechnology, sc-29528) according to manufacturer's instructions. siRNAs targeting human SLC45A4 (SCBT, sc-77846) and non-targeting Scramble controls (SCBT, sc-37007) were obtained from SCBT and re-suspended at 10 µM in RNAse free water. 4 µl siRNA (20 nM) was first mixed with 100 µl Opti-MEM (Gibco, 31985070), then mixed with 4 µl siRNA Transfection Reagent contained in another 100 µl Opti-MEM. The mixture was incubated at room temperature for 30 min and mixed with 800 µl Opti-MEM. Cells were washed once with 2 ml Opti-MEM, then covered with 1 ml siRNA mixture and incubated for 6 h at 37 °C, 5% $CO_2$. After the incubation, 1 ml of RPMI 1640 media with 20% FBS and 2× Penicillin-Streptomycin were added. Media was replaced the next day with the one containing stable isotope tracers if necessary.

### Construction of stable SLC45A4 knock down and knock off cell lines
CRISPR/Cas9 was used to interfere single (knock-down) or double allele (knockout) of SLC45A4 genomic sequence by the Genome Editing Shared Resource of Rutgers Cancer Institute.

## Stable Isotope Tracing

Cells were seeded in new 12-well plates at the density of $2 \times 10^5$/ ml. When the cells reached about 70% confluence, the media was replaced with SILAC RPMI 1640 Flex Media (Gibco, A2494201) supplemented with 11.11 mM D-glucose or, 0.22 mM L-lysine, 2.05 mM L-glutamine or $^{13}C_5$-glutamine (CLM-1822-H), 1.15 mM L-arginine or $^{13}C_6$-arginine (CLM-2265-H), 10% FBS and 1x Penicillin-Streptomycin. When appropriate, the following substrates were also added: 200 μM $^2H_6$-ornithine (DLM-2969), 5 μM (Fig. 3f, g) or 20 μM $^{13}C_4$-putrescine (Fig. 4f–h) (CLM-6574) and 20 μM GABA or $^{13}C_4$-GABA (CLM-8666). For subsequent labeling experiments involving $^2H_6$ ornithine, FBS was substituted with iFBS and 200 μM $^2H_6$-ornithine were added. All cells were cultured with isotope tracers for 24 h and metabolite extraction was performed.

## Proteoliposome preparation

The SLC45A4 and OMP25 proteoliposomes were synthesized using ProteoLiposome BD Kit (Cell-Free Science, Japan)[61,62] following the manufacturer's protocol. The synthesized proteoliposomes were centrifuged at $16,000 \times g$ for 15 min at 4 °C and were washed and resuspended with 1X HEPES-buffered saline (pH 7.4) (Thermo Scientific, J67502) containing 25 mM HEPES and 150 mM NaCl. This washing step was repeated three times to remove the translation buffer. The proteoliposomes were then resuspended in 1 mL of 1X HEPES-Saline buffer (pH 7.4), and stored at −80 °C.

## Proteoliposome size and concentration measurements

The SLC45A4- and OMP25-containing proteoliposomes were subjected to extrusion using a mini-extruder (Avanti Research, 610000) involving 10 cycles through a 400 nm Nuclepore Track-Etch membrane (Cytiva, 10417104), followed by 40 cycles through a 200 nm membrane (Cytiva, 10417004).

The size and concentration of extruded proteoliposomes were analyzed using a Tunable Resistive Plus Sensing (TRPS) system (IZON Science, Exoid) equipped with an 85–500 nm nanopore ((IZON Science, NP200). Prior to analysis, proteoliposomes were diluted 2000-fold in 1X PBS. The TRPS system was calibrated using a standard calibration particle with a mean size of 195 nm and concentration of $9.4 \times 10^{11}$ particles/mL (IZON Science, CPC200), diluted 500-fold in 1X PBS. Data acquisition and analysis were performed using the Izon Data Suite software.

## $^3$H uptake assay

150 μl of the prepared proteoliposome were mixed with 5 μL of $^3$H-labeled putrescine (Putrescine [2,3-$^3$H] dihydrochloride) (American Radiolabeled Chemicals, ART0279-1 mCi) and incubated at room temperature for 1 h. Excess putrescine was then removed by passing the samples through a size exclusion (optimum isolation size of 35–400 nm)-desalting column (IZON Science, qEV single 35 nm Gen 2) pre-equilibrated with 9 mL of 1X HEPES-Saline buffer (pH 7.4). The eluted fraction was collected in a total volume of 850 μL and a 425 μL aliquot of the elution was mixed with 10 ml of EcoLite Liquid Scintillation Cocktail (MPbio, 01882475-CF) and vortexed for 5–10 s. Radioactivity was measured using a Liquid Scintillation Counter (Beckman, LS6500), and the obtained counts per minute (CPM) were normalized to the calculated particle numbers in 150 μl of proteoliposomes.

## Metabolite extraction

For whole cell metabolite extraction, $0.5 \times 10^6$ cells in the culture were washed twice with PBS and extracted with 0.5 ml ice-cold 40:40:20 (methanol:acetonitrile:water) solution with 0.5% formic acid. The cells were scraped off the plate followed by incubation on ice for 10 min, and neutralized by the addition of 25 μl 15% (m/v) $NH_4HCO_3$ solution. The extracts were transferred to a 1.5 ml tube. For conditioned media metabolite extraction, 5 μl of culture media was mixed with 500 μl of ice-cold 40:40:20 (methanol:acetonitrile:water) solution with 0.5% formic acid and neutralized by the addition of 25 μl 15% (m/v) $NH_4HCO_3$ solution. The whole cell or media extracts were then centrifuged at 4 °C and $16,000 \times g$ for 10 min and transferred to a clean tube and stored at −80 °C until analysis.

## Metabolomic LC-MS analysis

HILIC separation was performed on a Vanquish Horizon UHPLC system (ThermoFisher) with an XBridge BEH Amide column (150 mm × 2.1 mm, 2.5 μm particle size, Waters) using a gradient of solvent A (95%:5% H2O:acetonitrile with 20 mM acetic acid, 40 mM ammonium hydroxide, pH 9.4) and solvent B (20%:80% H2O:acetonitrile with 20 mM acetic acid, 40 mM ammonium hydroxide, pH 9.4). The gradient was 0 min, 100% B; 3 min, 100% B; 3.2 min, 90% B; 6.2 min, 90% B; 6.5 min, 80% B; 10.5 min, 80% B; 10.7 min, 70% B; 13.5 min, 70% B; 13.7 min, 45% B; 16 min, 45% B; 16.5 min, 100% B; and 22 min, 100% B. The flow rate was 300 μl/min with 5 μl injection volume. The column temperature was 25 °C and the autosampler temperature was 4 °C. MS scans were obtained on a Q Exactive PLUS mass spectrometer (ThermoFisher) in both negative and positive ion modes with a resolution of 70,000 at m/z 200, in addition to an automatic gain control target of $3 \times 10^6$ and m/z scan range of 72 to 1000. Metabolite data was obtained using the El-MAVEN software package[63] (mass accuracy window: 5 ppm). All isotope natural abundance corrections were done using AccuCor[64,65].

## GABA quantitation

A549, H1299 and HepG2 cells were cultured in 6-well plates in regular media until reaching confluency. Triplicates were trypsinized and counted to evaluate average cell number while other triplicates were used for metabolomic extraction as described above. $^{13}C_4$-GABA was spiked into cellular and media extracts at the concentration of 1 μM before performing metabolomic LC-MS analysis as described above. Ratio between endogenous $^{12}C_4$-GABA and exogenous $^{13}C_4$-GABA was calculated to determine the concentration of GABA in the samples.

## Immunofluorescence

A549 WT cells or transiently overexpressing SLC45A4-3HA were seeded on poly-L-lysine coated glass coverslips (Neuvitro, GG18PLL) at the density of $2\times10^5$ / ml. When the cells reached about 70% confluence, cells were washed twice with PBS then were fixed using 4% paraformaldehyde (1% paraformaldehyde for E-cadherin detection) (Thermo Scientific Chemicals, J61899AK) in PBS for 15 min at room temperature. Fixed cells were then washed three times in PBS, permeabilized with Triton X-100 0.2% in PBS 1X for 20 min and blocked in PBS 1X containing 2% Normal Goat Serum (Invitrogen, 31873), 0.02% Triton X-100, 0.1% BSA for 30 min at room temperature (PBS 1X containing 1% BSA for 30 min at 37 °C for E-cadherin detection) (Blocking buffer). Primary antibodies were diluted in blocking buffer and incubated with cells overnight at 4 °C: Mouse anti-HA antibody (CST 2367, 1:800), rabbit anti-SLC45A4 (Invitrogen #PA5-54711, 1:1000), mouse anti-ABCD3 (Santa Cruz sc-514728, 1:800), mouse anti E-cadherin (CST 14472S, 1:200), mouse anti-Giantin (Abcam ab37266, 1:100) and mouse anti-Calnexin (Santa Cruz sc-23954, 1:100). After washing twice with TBS Tween-20 0.1% and once with blocking buffer, secondary antibodies were diluted in blocking buffer and incubated with cells for 1 h at room temperature in the dark. Anti-mouse IgG Alexa Fluor 594, (CST, 8890S), Alexa Fluor™ 647 Phalloidin (Invitrogen A22287 1:600), Alexa Fluor™ 546 goat anti-rabbit (Invitrogen A11010, 1:500) and goat anti-Rabbit IgG-Alexa Fluor® 488 (CST 4412S, 1:500) were used when appropriate. Cells were washed three times with PBS containing 0.2% Triton X-100. Coverslip were mounted with Fluoroshield with DAPI (Sigma, F6057). Images were obtained using an LSM-700 confocal microscope setup that includes 405, 488 and 555 nm lasers and 63× objective. Image analysis was performed on ZEN 3.6 software (Zeiss).

## RNA analysis

Total RNA was obtained from cell culture using the RNeasy Plus Mini Kit (Qiagen, 74004) following supplier instructions. Complementary DNA (cDNA) was generated from 1 μg total RNA using the iScript cDNA Synthesis kit (Bio-Rad, 1708841). Primers were designed with NCBI Primer-BLAST. The primer sequences for detecting *SLC45A4* are 5′-GATGGCCATGTTTCCCAACG-3′(forward) and 5′-GGCTGTGGTGGA TGTACTGC-3′ (reverse). The primer sequences for detecting *ACTB* are 5′- GACCTGACTGACTACCTCAT-3′(forward) and 5′-TCTCCTTAATGT CACGCACG-3′ (reverse) Quantitative PCR (qPCR) reactions were conducted using the SYBR green PCR master mix (Bio-Rad, 1725121) in CFX96 Touch Real-Time PCR Detection System (Bio-Rad). CT values were normalized to *ACTB* mRNA and interpreted as fold changes to the Ctr or WT cell group.

## Peroxisome Isolation by HA-IP Immunoprecipitation

Cells stably expressing 3xHA-eGFP-PEX26 construct were washed and scraped from two 150 mm culture plates (~ 30 million cells) in 2 ml pre-chilled PBS and centrifuged. Throughout the procedure, the cells and lysates were maintained on ice. The cell pellet was suspended in 1 ml PBS and 5 μl of the cell suspension was kept as whole cell (WC) input (0.05% input). The cells were pelleted at 800 g for 5 min then resuspended in 1 ml of hypotonic buffer (VWR, 97063-130) and incubated on ice for 15 min. The cells were subjected to stepwise centrifugation steps: 5 min at 1000 g (to pellet nucleus), 5 min at 2000 g (to pellet heavy mitochondria), 5 min at 3000 g (to separate light mitochondria), changing tubes at every centrifugation. The resulting lysate was mixed with 180 μl of anti-HA magnetic beads (Thermo, 88837) and incubated for 30 min at 4 °C. The immune-precipitate was washed 2X with wash buffer (1 mM EDTA, 50 mM Tris-HCl, pH 7.5 and 150 mM NaCl) and once with wash buffer supplemented with 0.05% NP-40. To extract bead-bound proteins, the beads were incubated with lysis buffer (1 mM EDTA, 50 mM Tris-HCl, pH 7.5 and 150 mM NaCl, 1% NP-40) on ice for 30 min. The lysate was subjected to SDS-PAGE electrophoresis and transferred to nitrocellulose membrane (See below "*Protein extraction and western blotting*").

## Peroxisome isolation by density gradient centrifugation

Endogenous peroxisomes were isolated from WT A549 cells using Peroxisome Isolation kit (Sigma Aldrich, PEROX1-1KT). Briefly, about 300 million WT A549 cells were washed with PBS, scraped and centrifuged at 800 g for 5 min to obtain a packed cell volume of 2.5 ml, to which 6.2 ml extraction buffer (5 mM MOPS pH 7.65, 0.25 M sucrose, 1 mM EDTA and 0.1% ethanol) was added. The cells were homogenized in a 10 ml Potter-Elvehjem homogenizer using 100 gentle douncing strokes. The cell suspension was centrifuged at 1000 g and the supernatants obtained from subsequent steps were stepwise centrifuged at 2000 g for 10 min, 3000 g for 10 min, and 25,000 g for 40 min. The pellet obtained at each step were extracted in RIPA buffer equivalent to 2.5x pellet size and kept for protein quantitation and SDS-PAGE. The pellet obtained from the last centrifugation step was layered in OptiPrep™ density gradient medium containing iodixanol (60% w/v) and centrifuged for 17 h at 100,000 g in a Type 41Ti, swinging bucket rotor of an Optima™ XL-80K Backman Coulter Ultracentrifuge. Post-centrifugation, different layers were collected, washed in PBS and concentrated at 21,000 g for 60 min. The pellets thus obtained were reconstituted in RIPA buffer, the lysate protein was quantified using Bradford method and the lysates were subjected to SDS-PAGE.

## Protein extraction and western blotting

One million cells were washed once with PBS and extracted with 100 μl protein extraction RIPA buffer and protease inhibitors from Santa Cruz Biotechnology (SCTB, sc-24948). The extracts were incubated 10 min on ice and sonicated 5 s for three times. The sonicated extracts were mixed with 4× Laemmli loading dye containing 5% 2-mercaptoethanol and incubated at room temperature for 10 min without boiling. 10 μg of protein samples were loaded per lane and electrophoresed on Mini-PROTEAN® TGX™ Precast Protein Gels (BioRad) and transferred to 0.2 μm nitrocellulose membrane using semi dry transfer system (BioRad). Membranes were incubated with blocking buffer (5% milk in Tris-buffered saline (TBS) containing 0.1% (v/v) Tween-20 (TBST)) for 1 h at room temperature and incubated with primary antibodies at 4 °C overnight. Primary antibodies were diluted in the blocking buffer: Citrate Synthase (CS) (CST 14309, 1:1000); β-Actin (ACTB) (CST 4967, 1:1000); SLC45A4 (Genescript R11554, 1:3000), E-cadherin (CST 3195S, 1:3000), Catalase (CST 12980S, 1:1000), ABCD3 (SCBT 514728, 1:3000). After three washes with TBST, membranes were incubated 1 h at room temperature with the secondary antibody diluted in blocking buffer: anti-rabbit (CST 7074, 1:2000) or anti-mouse (Invitrogen 62-6520, 1:2000). After three washes with TBST, membranes were incubated with HRP substrate (Millipore 221543) and imaged using ChemiDoc touch imaging system (Bio-Rad).

## Statistical analysis

Statistical analyses were performed using Excel 2016 for Student's t-test and R/R-Studio for ANOVA.

## Reporting summary

Further information on research design is available in the Nature Portfolio Reporting Summary linked to this article.

## Data availability

The MS data generated in this study and used for Fig. 1e, f, 3c, d, 4d. have been deposited in the Metabolomic Workbench database under project ID PR002513. https://doi.org/10.21228/M8T536. Source data are provided with this paper.

## Code availability

Computer code is available from Zenodo https://doi.org/10.5281/zenodo.15832229.

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

## Acknowledgements

This work was supported by NIH grant R01GM149664, the Rutgers Health Busch Biomedical grant, and the RWJMS DOM Seed Fund. X.S., C.C., N.R.D., D.K., and J.A. are supported by NIH R01GM149664. C.C. was supported in part by the Rutgers Health Busch Biomedical grant. This work was also supported in part by the Rutgers Cancer Institute of New Jersey Metabolomics Shared Resource (NCI-CCSG P30CA072720-5923) and Genome Editing Shared Resource (NCI-CCSG P30CA072720-5922). We thank Dr. Maria Elena Diaz-Rubio and Dr. Eric Chiles in Su Lab for their technical assistance on the LC-MS experiments and their comments on this manuscript.

## Author contributions

X.S. conceived this project. C.C., Y.W., and X.S. designed the study. C.C, Y.W., J.A., N.R.D., D.K., X.S. performed experiments and data analyses. X.S. supervised the whole study. C.C., Y.W. and X.S. wrote the original and revised manuscript.

## Competing interests

The authors declare no competing interests.
