## [Transparent Peer Review file · Nature Communications]

SLC45A4 encodes a peroxisomal putrescine transporter that promotes GABA de novo synthesis

Corresponding Author: Dr Xiaoyang Su

Version 0:

Reviewer comments:

Reviewer #1

(Remarks to the Author)

SLC45A4 encodes a mitochondrial putrescine transporter that promotes GABA de novo synthesis

The paper is interesting and well-written and intuitive, with a good flow. However, there are some concerns that need to be addressed.

Major

The authors suggest that the increase in GABA observed in the conundrum might result from external putrescine being converted to GABA by DAO and then taken up by the cells. However, this interpretation requires caution. Could the authors not have tested GABA transporter blockers to explore this possibility further? Additionally, it would be important to validate the contribution of endogenous enzymes, perhaps through the use of DAO shRNA or knockout cells, to ensure that the increase in GABA is not solely due to serum DAO activity, as the authors seem to suggest. Aminoguanidine, while effective, targets not just DAO but also SSAO, NOS, and other enzymes, which complicates the interpretation of its effects.

Changes in putrescine levels might induce mitochondrial alterations (M Maccarrone et al., 2001), potentially leading to an adaptive response of SLC45A4 to putrescine. Since the authors propose that SLC45A4 is a mitochondrial putrescine transporter, the authors should address potential structural or functional changes in mitochondria in response to putrescine change.

In Figure 3, the authors mentioned that SLC45A4 KO is independent of extracellular putrescine oxidation. Authors seem to assume that putrescine is not permeable to the plasma membrane. Is there any report or data on putrescine being impermeable to the plasma membrane and oxidized in extracellular space? In addition, there should be a few ¹³C putrescine and increased ¹³C 4-aminobutanol in the cytosol. Did authors measure ¹³C putrescine and ¹³C 4-aminobutanol level when treated the ¹³C putrescine at the extracellular space?

In Figure 4d, the authors showed that ODC1 overexpression cannot rescue the GABA level. In the ODC1 overexpression, the intracellular putrescine level will increase, but increased intracellular putrescine cannot be oxidized in the SLC45A4 KO. This data is inconsistent with Figure 3C, which showed that 2H-ornithine-induced intracellular putrescine cannot be oxidized in the SLC45A4 KO. How can there be intracellular ¹³C GABA in the SLC45A4 KO?

Supplementary Figure 4: The authors firmly state that SLC45A4 is not a sucrose transporter based on indirect evidence. However, cells may preferentially utilize glucose when both glucose and sucrose are present, possibly reducing the observed sucrose transport activity. What happens to sucrose incorporation when glucose is absent, and SLC45A4 is manipulated (e.g., KO or KD)?

Overall, I can see that SLC45A4 is essential for putrescine transport from cytosol to mitochondria. However, these results can also be explained if SLC45A4 supports putrescine transport. So, it might be too early to say SLC45A4 is the putrescine transporter. I am not entirely convinced that SLC45A4 is definitively a putrescine transporter or, rather, a critical protein that assists another putrescine transporter. Authors should tone down their claims. Could structural simulations using tools like

AlphaFold help clarify this?

Minor

In Figure 1b, the GABA and SLC45A4 RNA levels for HepG2 are nearly 0, and the authors claim that the low level of GABA is due to the low expression of SLC45A4. However, the GABA level in Figure C is around 1, more significant than the GABA level of H1299 in Figure 1b, implying that there is SLC45A4 expression in the HepG2 cell line inconsistent with your data. Is there any difference in the experimental conditions in Figures 1b and 1c?

Line 102-103: "Meanwhile, we observed very little ¹³C labeling (<1%) in GABA when ¹³C6-glucose was used as the tracer." This statement lacks a referred figure or reference.

Why is ¹³C4 GABA decreased in the SLC45A4 KD condition?

The plot in Figure 4g is challenging to interpret and could benefit from a more intuitive presentation.

It might also be worth mentioning both neuronal and astrocytic GABA synthesis from putrescine, as they play a significant role in brain GABA synthesis (Koh et al., 2023). This could be a valuable addition to the discussion section.

Reviewer #2

(Remarks to the Author)

This is an interesting manuscript from Colson et al that explores the biochemical function of SLC45A4, a poorly studied solute carrier. The main claims are that 1) SLC45A4 encodes a mitochondrial putrescine transporter; and 2) by regulating the arginine/ornithine/putrescine pathway, SLC45A4 controls GABA synthesis.

Overall the novelty of this manuscript is high. Previously, the biochemical function of SLC45A4 was thought to be in the transport of sugars. Linking this transporter to GABA pathway metabolism is interesting and has not been previously described. Also the approach to associate metabolites and gene expression to find the SLC45A4-GABA association interesting. Lastly there is strong experimental data in cells to support that genetic manipulation (overexpression or knockout) of SLC45A4 modulates intracellular levels of GABA (claim #2).

However, there are also moderate to major experimental weaknesses, especially with claim #1. In its current form, these weaknesses require major revision. Nevertheless, if adequately addressed, such a revision would dramatically strengthen the conclusions of this very novel manuscript.

Major

1. There is no direct in vitro data to show that SLC45A4 is a putrescine transporter. Many of the cellular tracing experiments in cells could be complex and indirect, as the authors have abundantly demonstrated. Furthermore, there is no guarantee that the inhibitors used in Fig 4 are specific, and potential non-specific effects of these inhibitors may affect data interpretation. The authors should attempt to perform direct reconstitution and transport experiments, as was previously described in Bartolke Biochem J 2014, to determine if reconstituted SLC45A4 is sufficient to confer putrescine transport in yeast or proteoliposomes. If direct reconstitution cannot be established, then a direct claim of a specific transporter activity cannot be made; rather, the claims should be modified eg "SLC45A4 is necessary for GABA synthesis in cells" etc.
2. The data showing that SLC45A4 is mitochondrial is weak. First, the GFP tagged version in Fig 5 does not seem to co-localize with mitochondria. Second, the fractionation performed in (b) is complicated since lysosomes and other organelles can also co-fractionate with mitochondria. Third, SLC45A4 is not in mitocarta or other mitochondrial proteome compendiums. Fourth, SLC45A4 does not have an obvious mitochondrial targeting signal. My speculation would be that this transporter is actually in another compartment?

Minor:

3. As negative controls, it would also be informative to understand whether overexpression or knockdown of the other SLC45A1/2/3 members altered GABA levels in cells.
4. The relevant cell type for GABA biosynthesis is not A549, H1299, or HEP62 cells, but rather GABAergic neurons. Does genetic manipulation of SLC45A4 in neurons also modulate GABA levels?
5. The data in Supplementary Fig 1 showing t-stat values between the 225 metabolites and 464 SLC genes is interesting. To help facilitate open data sharing, such data should be supplied as raw data in a searchable excel sheet.

Version 1:

Reviewer comments:

Reviewer #2

(Remarks to the Author)

The authors have satisfactorily addressed my questions

Reviewer #3

(Remarks to the Author)

The manuscript by Colson C et al. reveals a novel function of the solute carrier protein SLC45A, which facilitates the transport of putrescine across peroxisome membrane, subsequently enhancing GABA de novo synthesis via the AOP pathway. However, the study's generalizability is limited due to the experimental data being generated solely from two lung cancer cell lines. Several critical questions remain unanswered regarding the impact of this enhanced GABA de novo synthesis by SLC45A on cellular physiology. Several concerns need to be addressed.

Major points:

1. Lung cancer cell lines, including A549, which was used in this study appear to express GAD1 (doi: 10.1038/s41556-021-00820-9) and have been reported to synthesize and release GABA. How do the authors exclude the contribution from GAD1 to GABA production?
2. Although LC-MS analysis can quantify GABA levels, it does not seem to allow direct comparison of GABA concentrations across different experimental runs (e.g. Fig 1b, right panel, HepG2 vs Fig 1c HepG2 Ctr). It would be useful to validate the findings using alternative methods, such as GABA ELISA or GABA-sensor or electrophysiology with GABAAR receptor-expressed cells, to determine the intracellular and extracellular GABA concentrations.
3. The authors propose the conversion of external putrescine to GABA, which is then transported into the cells. However, several proteins, in addition to canonical GABA transporters, could mediate GABA transport across the membrane, such as TauT, BGT, BEST, VRAC. The authors should investigate this possibility. Notably, a GAT2/BGT1 inhibitor is also available (<https://doi.org/10.1016/j.neuint.2005.12.031>).
4. The study was conducted in only two lung cancer cell lines. It would be valuable to assess GABA levels in other cancer cell lines or primary cells with high SLC45A expression to further validate the findings.
5. In Figure 2c, is it correct that ¹³C-arginine labeled approximately 60% GABA with ¹³C in HepG2 cells?
6. Several protein bands for SLC45A4 are shown in Figure 3b. The specificity of this antibody should be further validated using other cell or tissue lysate with varying SLC45A4 expression levels..
7. On page 5, line 192, it is stated: "All the DAO inhibitors showed inhibition of GABA production from 2H-ornithine, but neither of the MAO inhibitors did (Figure 4e)". However, did clorgyline significantly reduce 2H-GABA, as shown in figure 4e right panel?
8. In figure 5, the staining of SLC45A4 appears heterogenous, with not all cells expressing SLC45A (especially in the right panel). Immuno-EM would help confirm the subcellular localization of SLC45A4..
9. Both cell lines used in this study express functional GABA receptors, and exogenous GABA has been shown to affect their proliferation (doi: 10.1186/1479-5876-11-102, <https://doi.org/10.1038/srep32756>), it would be interesting to investigate whether de novo synthesized GABA by SLC45A4 acts its own receptor in an autocrine manner.

Minor points:

1. The ¹³C-GABA concentration in Figure 1d is reported as 20 uM, which differs from the legend (5 uM). Please clarify.
2. Page 3, line 116, ¹³C-GABA is significantly reduced in A549KD cells (Fig 1e right panel).
3. For improved statistical reliability, it would be preferable to include at least 5 independent replicates rather than 3.
4. Figure 4g, are the groups with 100 nM aminoguanidine statistically significant? Please clarify
5. Page 6, line 225, left panel; line 228, right panel.

Version 2:

Reviewer comments:

Reviewer #3

(Remarks to the Author)

Thank you for thoroughly addressing my comments! The revision as enhanced the clarity and scientific quality of the paper.

Dear Reviewers,

We are submitting the revision of our manuscript NCOMMS-24-48063 for your review. The title of this manuscript has been changed to “*SLC45A4* encodes a peroxisomal putrescine transporter that promotes GABA *de novo* synthesis”. This is a critical change we made in the revised manuscript, based on our new experimental evidence. We will begin our response letter with an explanation of this new conclusion. We apologize for the confusion we may have caused in the first submission. We sincerely appreciate the reviewers’ suggestions and feel that the resulting changes have substantially improved the quality of this manuscript.

SUBCELLULAR LOCALIZATION OF SLC45A4

A key change in the revised manuscript is that we observed endogenous SLC45A4 localized to peroxisome rather than mitochondria. In our previous submission, our conclusion is based on the results from the immunofluorescence experiments in which we over-expressed SLC45A4 with a GFP- or an HA-tag. While these epitope tags enabled an easy detection of SLC45A4, the over-expression may have resulted in an excess amount of SLC45A4 protein and perhaps an abnormal localization. To address this issue, we newly obtained an SLC45A4-specific antibody (Invitrogen, Cat # PA5-54711). We first verified the specificity of this antibody. The green (HA) and red (SLC45A4) signals perfectly co-localized with our over-expressed HA-tagged SLC45A4 protein (Supplementary Figure 8). Next, we stained the cells with Invitrogen anti-SLC45A4 antibody to detect the localization of the endogenous SLC45A4 protein. Our results show the signals are strongly co-localized with ABCD3, which is a peroxisomal transporter of long-chain fatty acids and is frequently used as a peroxisomal marker (**Figure 5a.**). Given this new observation, we have revised our manuscript to reflect the peroxisomal localization of SLC45A4. The section on mitochondrial localization has been removed.

Supplementary Figure 8. Anti-SLC45A4 (Invitrogen) specifically detects overexpressed SLC45A4 with an HA-tag. A549 WT cells transiently overexpressed SLC45A4-HA and were stained for both HA (green) and SLC45A4 (red).

Figure 5. SLC45A4 encodes a peroxisomal putrescine transporter. a. Immunofluorescent detection of SLC45A4 subcellular localization. SLC45A4 endogenous, green; ABCD3, peroxisome marker, red.

Supplementary Figure 9. Immunofluorescent detection of SLC45A4 subcellular localization. SLC45A4 endogenous, green; organellar markers, Giantin (Golgi apparatus), Calnexin (Endoplasmic reticulum), and Mitotracker (Mitochondria) red.

We have made the corresponding changes in the main text lines 221-230 and in Figure. 5a.

First, we tested the specificity of a commercial SLC45A4-targeting antibody by transiently overexpressing SLC45A4 with an HA-tag and observed perfect colocalization between green signals (anti-HA) and red signals (anti-SLC45A4) (Supp. Figure. 8). Therefore, we used this antibody to detect endogenous SLC45A4 transporter in A549 cells. SLC45A4 (green signal) is clearly detected around the cell nucleus, without overlapping actin signal (Figure. 5a., top panel), suggesting organellar localization rather than plasma membrane localization. Moreover, SLC45A4 signal strongly co-localized with ABCD3, which is a peroxisomal transporter of long-chain fatty acids and is frequently used as a peroxisomal marker (Figure. 5a, lower panel). In addition, we observed no overlap between SLC45A4 signals and Golgi marker, Giantin; endoplasmic reticulum marker, Calnexin or mitochondrial marker, Mitotracker (Supp. Figure. 9).

REVIEWER COMMENTS

Reviewer #1

Reviewer 1 commented that this paper is interesting, well-written and intuitive. Meanwhile, the reviewer has some concerns. These comments are quite relevant and constructive. We want to take this opportunity to thank the review. And our responses are listed below.

Major critiques

Comment 1: *The authors suggest that the increase in GABA observed in the conundrum might result from external putrescine being converted to GABA by DAO and then taken up by the cells. However, this interpretation requires caution. Could the authors not have tested GABA transporter blockers to explore this possibility further?*

Response 1: We thank the reviewer for the remark on the role of canonical GABA transporter (GAT1-SLC6A1/GAT2-SLC6A13/GAT3-SLC6A11) in SLC45A4-mediated GABA production. While these GABA transporters have critically important roles in other cells, in our cells they are barely expressed (See below Figure R1a). We believe the expression of GATs is mostly limited to neurons and astrocytes where they contribute to GABA re-uptake.

GAT2 (SLC6A13) has the highest expression levels among the GATs, based on the CCLE database. We also verified the expression level of GAT2 in our cells (See below Figure R1b). Unfortunately, GAT2 has no known potent inhibitor because it's a low affinity/high selectivity transporter. Few inhibitors are known to inhibit more than 50% of GABA transport and only at high concentrations such as mM and 500 μ M (PMID: 22932902). Therefore, we did not perform GAT2 inhibition experiments.

While we believe GABA is ultimately made intracellularly, possibly depending on the activities of cellular ALDHs, we admit we don't have direct evidence at this moment whether the GABA is made extracellularly and is taken up by GATs. Our future work will focus on this.

Figure R1. Endogenous expression of canonical GABA transporters. a. CCLE database RPKM for known GABA transporters and SLC45A4 (far right) **b,** RT-qPCR comparison of relative mRNA levels SLC45A4 and SLC6A13-GAT2 in A549 (left) and H1299 (right) cells.

Comment 2: Additionally, it would be important to validate the contribution of endogenous enzymes, perhaps through the use of DAO shRNA or knockout cells, to ensure that the increase in GABA is not solely due to serum DAO activity, as the authors seem to suggest. Aminoguanidine, while effective, targets not just DAO but also SSAO, NOS, and other enzymes, which complicates the interpretation of its effects.

Response 2: We thank the reviewer for the insightful remark on the role of endogenous and exogenous DAOs. The contribution of extracellular DAOs is indeed elusive. We totally agree with Reviewer 1 that this is a key question that needs to be addressed. Based on the CCLE database and our own experimental data, AOC2 and AOC3 are the two major DAOs in A549 and H1299 cells. While we had an effective KD of AOC2 and AOC3 (data not shown), we did not see its impact on GABA levels in our cellular models. It is possible that the decreased cellular DAO activity is compensated by putrescine export and oxidation.

To further characterize the intracellular putrescine oxidation for GABA production and to show that putrescine oxidation cannot be completely attributed to extracellular DAOs, we added two experiments. First, we repeated the aminoguanidine (AG) treatment with lower doses. We agree with the reviewer that AG may have multiple targets. We used 100 nM and 10 nM instead of 1 μ M to minimize its side-effects. Our results show a higher dose of AG decreased the GABA levels more, but increased the ratio of $^2\text{H}/^{13}\text{C}$ GABA (**Figures 4f and 4g Supp Figure. 7a**). This result supports the partial intracellular oxidation of putrescine. If all putrescine, ^{13}C and ^2H , were oxidized exclusively extracellularly, the AG treatment should have no effect on the $^2\text{H}/^{13}\text{C}$ GABA ratio. Second, we cultured cells with regular dialyzed and heat-inactivated FBS

(dFBS) and with horse serum (Hs), respectively. The motivation for this experiment is that Hs has a much lower DAO activity compared to FBS (PMID 3106317, 8789157). Meanwhile, we supplemented the cells with 200 μM of $^2\text{H}_6$ -ornithine and 20 μM of $^{13}\text{C}_4$ -putrescine. Our results show that Hs significantly lowered total GABA level, suggesting that supplemented putrescine is most oxidized by the DAO from the serum. Nonetheless, the cells cultured in Hs have a higher $^2\text{H}/^{13}\text{C}$ GABA ratio, suggesting the production of ^2H -GABA is partially dependent on the intracellular DAOs (**Figure 4h. and Supp Figure 7b.**).

Figure 4f. Fractions of cellular GABA in A549 WT cells supplemented with $^2\text{H}_6$ -Ornithine and $^{13}\text{C}_4$ -Putrescine in the absence (0) or presence of 10 or 100 nM of aminoguanidine. **g.** Cellular ratio of ^2H -GABA/ $^{13}\text{C}_4$ -GABA in A549 WT cells supplemented with $^2\text{H}_6$ -Ornithine and $^{13}\text{C}_4$ -Putrescine in absence (0) or presence of 10 or 100 nM of aminoguanidine.

Supp Figure 7a: Cellular fraction of GABA in A549 cells in the absence (0) or presence of 10 or 100 nM of aminoguanidine.

Figure 4h. Cellular ratio of ^2H -GABA/ $^{13}\text{C}_4$ -GABA in A549 cells supplemented with $^2\text{H}_6$ -Ornithine and $^{13}\text{C}_4$ -Putrescine in regular serum (dFBS) or horse serum (Hs).

Supp Figure 7b. Cellular ratio of ^2H -GABA/ $^{13}\text{C}_4$ -GABA in H1299 cells supplemented with $^2\text{H}_6$ -Ornithine and $^{13}\text{C}_4$ -Putrescine in regular serum (dFBS) or horse serum (Hs).

Manuscript was edited on lines 198-208:

First, we supplemented the cells with $^2\text{H}_6$ -ornithine and $^{13}\text{C}_4$ -putrescine treated with 10 nM or 100 nM or without the DAO inhibitor, aminoguanidine. As previously shown in Figure 4e., aminoguanidine decreases total cellular GABA but $^2\text{H}/^{13}\text{C}$ ratio increases by 2 folds (Figure. 4f, Supp. Figure. 7a), indicating that GABA derived from extracellular putrescine (^{13}C fraction) is reduced (10% reduction, Figure. 4g) while GABA derived from intracellular putrescine (^2H fraction) is augmented (2 folds, Figure. 4g and Supp. Figure. 7a). Second, we cultured the cells under regular dialyzed heat inactivated FBS (dFBS) or Horse serum (Hs), which has been shown to have a much lower DAO activity.⁴⁷ Our results show that, similar to aminoguanidine, $^2\text{H}/^{13}\text{C}$ ratio of cellular GABA is strongly increased, from 3 to 15 folds (Figure. 4h and Supp. Figure. 7b). These results suggest that supplemented putrescine is oxidized extracellularly and that the production of ^2H -GABA depends, at least partially, on the intracellular DAOs.

Comment 3: *Changes in putrescine levels might induce mitochondrial alterations (M Maccarrone et al., 2001), potentially leading to an adaptive response of SLC45A4 to putrescine. Since the authors propose that SLC45A4 is a mitochondrial putrescine transporter, the authors should address potential structural or functional changes in mitochondria in response to putrescine change.*

Response 3: We thank the reviewer for this comment. We apologize again for the confusion we have made in the first submission. Our new results support the idea that SLC45A4 is a peroxisomal transporter rather than a mitochondrial transporter. Regarding the potential role of this transporter on peroxisomes, knowing that DAOs oxidize putrescine and generate H_2O_2 , we think it makes sense that this reaction happens in peroxisomes.

Comment 4: *In Figure 3, the authors mentioned that SLC45A4 KO is independent of extracellular putrescine oxidation. Authors seem to assume that putrescine is not permeable to the plasma membrane and oxidized in extracellular space? In addition, there should be a few ^{13}C putrescine and increased ^{13}C 4-aminobutanol in the cytosol. Did authors measure ^{13}C putrescine and ^{13}C 4-aminobutanol level when treated the ^{13}C putrescine at the extracellular space?*

Response 4: We thank the reviewer for this comment on putrescine oxidation. Putrescine can be oxidized to 4-aminobutanal, which may also undergo a spontaneous cyclization reaction to form the Schiff base 1-pyrroline. We have tried to look for both compounds on our LC-MS but didn't detect any. These compounds are not commercially available. We are unable to buy these compounds to validate our HILIC LC-MS detection. Our guess is these compounds may not be very stable, hindering our detection.

Despite the difficulties getting 1-pyrroline, we purchased 2-methyl-1-pyrroline (Sigma #381055) and tested whether it can be taken up by the cells. Indeed, we were able to detect 2-methyl-1-pyrroline in both the media and the cellular fraction. We believe 1-pyrroline can be taken up by the cells, possibly through diffusion, and can be further oxidized to GABA by ALDHs.

Figure R2. 2-methyl-1-pyrroline is able to enter the cell.

Meanwhile, putrescine can be directly taken up by the cells (*Polyamine transport is mediated by both endocytic and solute carrier transport mechanisms in the gastrointestinal tract*, Uemura et al., 2010, PMID 20522643). However, it appears that the extracellular putrescine oxidation is much faster than the putrescine uptake through endocytosis and SLC3A2. This is why extracellular putrescine can be oxidized to GABA despite SLC45A4 KO. We have not done a SLC3A2 overexpression experiment, which will be a future endeavor.

Comment 5: In Figure 4d, the authors showed that ODC1 overexpression cannot rescue the GABA level. In the ODC1 overexpression, the intracellular putrescine level will increase, but increased intracellular putrescine cannot be oxidized in the SLC45A4 KO. This data is inconsistent with Figure 3C, which showed that 2H-ornithine-induced intracellular putrescine cannot be oxidized in the SLC45A4 KO. How can there be intracellular ¹³C GABA in the SLC45A4 KO?

Response 5: We thank the reviewer for this insightful comment on ODC1 overexpression experiment. ODC1 overexpression increased both intracellular and extracellular putrescine concentrations (Supplementary Figure 7). We believe there is plasma membrane transporter that can export excess cellular putrescine to the extracellular space to get oxidized for GABA production. Importantly, SLC45A4 KO cells also have putrescine export activity, albeit slightly lower than WT cells. When ODC1 was overexpressed in SLC45A4 KO cells, extracellular putrescine, which is ²H-labeled derived from ²H-ornithine, also increased dramatically. Therefore, the ODC1 overexpression caused an increased production of ²H₆-GABA in

SLC45A4 KO cells. This result is actually consistent rather than contradictory to Figure 3c, which shows SLC45A4 dramatically decreased the production of $^2\text{H}_6$ -GABA from $^2\text{H}_6$ -Ornithine, but didn't totally eliminate it.

Manuscript was edited for these results on lines 179-185:

It is also noteworthy that *ODC1* overexpression increased the extracellular putrescine concentration in both WT and SLC45A4 KO cells (Supp Figure 6). Therefore, the increase of GABA in *SLC45A4* KO after *ODC1* over-expression could come from the oxidation of extracellular putrescine, which does not depend on SLC45A4. Overall, the results from putrescine supplementation and *ODC1* overexpression suggest SLC45A4 is required for cellular putrescine oxidation, and it hinted to us to consider how the intracellular and extracellular putrescine is differentially oxidized.

Supplementary Figure 7. ODC1 overexpression increases a. intracellular and b. extracellular putrescine. Cellular (top panel) and media (bottom panel) levels of putrescine in A549 (left) and H1299 (right) WT and their SLC45A4-KO cells (KO1 & KO2) overexpressing empty vector (Ctr) or human-ODC1 (OE-hODC1).

Comment 6: *Supplementary Figure 4: The authors firmly state that SLC45A4 is not a sucrose transporter based on indirect evidence. However, cells may preferentially utilize glucose when*

both glucose and sucrose are present, possibly reducing the observed sucrose transport activity. What happens to sucrose incorporation when glucose is absent, and SLC45A4 is manipulated (e.g., KO or KD)?

Responses 6: We thank the reviewer for this remark on the possible function of SLC45A4 as a sucrose transporter. We agree with the reviewer that it is a common phenomenon that the presence of high concentration glucose may suppress the uptake and utilization of other carbon sources such as sucrose. Therefore, even if we observed no isotope dilution in glycolytic intermediates derived from ^{13}C -glucose, it may not fully reject the possibility of SLC45A4 having a sucrose activity when glucose is absent. We agree with the reviewer on this point, and we have changed the text accordingly. However, our point is that the GABA-producing activity has nothing to do with the possible sucrose-transporting activity of SLC45A4, since this activity is robust under normal glucose concentration.

We have revised the manuscript with the following paragraph (Line 97-105).

A549 and H1299 cells were cultured in $^{13}\text{C}_6$ -glucose with or without supplemented ^{12}C -sucrose. If sucrose is catabolized by these cells, it should be incorporated into the glycolytic and the TCA cycle intermediates, and therefore decrease the ^{13}C enrichment in these metabolites. Our observations did not support the sucrose-transporting activity. While we observed a robust ^{13}C labeling in glycolysis and TCA cycle intermediates, the addition of sucrose has no negative impact on them, suggesting sucrose is not catabolized in A549 and H1299 cells with the basal expression of SLC45A4 (Supp Figure 4). While our experiment does not completely rule out the sucrose-transporting activity of SLC45A4 in the absence of glucose, the regulation of GABA synthesis is likely not due to the sucrose-transporting activity.

Comment 7: *Overall, I can see that SLC45A4 is essential for putrescine transport from cytosol to mitochondria. However, these results can also be explained if SLC45A4 supports putrescine transport. So, it might be too early to say SLC45A4 is the putrescine transporter. I am not entirely convinced that SLC45A4 is definitively a putrescine transporter or, rather, a critical protein that assists another putrescine transporter. Authors should tone down their claims. Could structural simulations using tools like AlphaFold help clarify this?*

Responses 7: We thank the reviewer for this remark on the direct evidence for putrescine transporting activity of SLC45A4. We used a wheat-germ cell-free translation system (CellFree Sciences, Ehime, JAPAN) to make reconstituted proteoliposomes containing SLC45A4. Our results show SLC45A4-containing proteoliposome has higher putrescine uptake activity compared to the control proteoliposomes, which contains a chimeric membrane protein eGFP-OMP25.

Manuscript was edited on lines 232-236

Finally, we confirmed the putrescine-transporting activity of SLC45A4. We used a wheat-germ cell-free translation system to make reconstituted proteoliposomes containing SLC45A4 or a chimeric membrane protein eGFP-OMP25 (Figure. 5b). We performed a ^3H -putrescine uptake assay and our results show that SLC45A4-containing proteoliposome has significantly higher putrescine uptake activity compared to the control proteoliposomes (Figure. 5b).

Figure 5b. SLC45A4-containing proteoliposomes (SLC45A4) and Control (Ctr, OMP25-containing proteoliposomes) in presence of ³H-putrescine.

Minor critiques

Comment 8: *In Figure 1b, the GABA and SLC45A4 RNA levels for HepG2 are nearly 0, and the authors claim that the low level of GABA is due to the low expression of SLC45A4. However, the GABA level in Figure C is around 1, more significant than the GABA level of H1299 in Figure 1b, implying that there is SLC45A4 expression in the HepG2 cell line inconsistent with your data. Is there any difference in the experimental conditions in Figures 1b and 1c?*

Responses 8: We thank the reviewer for this comment, we agree that it could be confusing. “Ion counts” is an arbitrary unit corresponding to the compound’s level of ionization within the experiment and not an absolute quantification of metabolite. Therefore, it is not possible to compare directly the ion counts of the same metabolite in two independent, even similar, experiments. Comparison can only be made between the compound’s level of two experimental conditions ionized within the same run.

Comment 9: *Line 102-103: "Meanwhile, we observed very little 13C labeling (<1%) in GABA when 13C6-glucose was used as the tracer." This statement lacks a referred figure or reference.*

Responses 9: We thank the reviewer’s comment that it was not clearly stated this statement referred also to the supplementary figure 4 mentioned above. We propose to add an extra figure displaying specifically GABA levels in this experiment as followed:

Supplementary Figure 4. SLC45A4 is not a sucrose transporter and GABA is not derived from glucose. **b.** Cellular GABA fractions in A549 and H1299 cells supplemented with $^{13}\text{C}_6$ -glucose in absence (-) or presence (+) of unlabeled sucrose (^{12}C -sucrose).

Comment 10: *Why is $^{13}\text{C}_4$ GABA decreased in the SLC45A4 KD condition?*

Responses 10: We want to thank the reviewer for this observation. We agree that a slight decrease is observed on $^{13}\text{C}_4$ -GABA cellular levels in A549 cell line KD for SLC45A4. However, this decrease is around 20% and only present in one cell line while decreased in unlabeled GABA ($^{12}\text{C}_4$ -GABA) is 50% in both cell lines. We mentioned it without the numbers at line 115-116: “The $^{13}\text{C}_4$ -GABA, which indicates GABA uptake, was only slightly changed in A549 KD cells but not in H1299 KD cells.”.

Comment 11: *The plot in Figure 4g is challenging to interpret and could benefit from a more intuitive presentation.*

Responses 11: We thank the reviewer for suggesting to clarify this part of Figure 4. Plots for Figures 4f, 4g and 4h were modified according to new results and the manuscript was edited lines 198-208 as mentioned previously for Comment 2.

Comment 12: *It might also be worth mentioning both neuronal and astrocytic GABA synthesis from putrescine, as they play a significant role in brain GABA synthesis (Koh et al., 2023). This could be a valuable addition to the discussion section.*

Responses 12: We thank the reviewer for this interesting comment and we agree that mentioning different modes of GABA synthesis in the central nervous system will improve the discussion. The latest review from Koh et al., 2024, Clinical and Translation Medicine (PMID: 38558537), describes non-neuronal cells in brain such as microglia and astrocytes able to produce and release GABA in small amount in the extracellular space in order to maintain a “GABA tone” to regulate neurons activity. This release is non-vesicular (different from neurons and synapses) and happens through channel-mediation. This GABA synthesis happens mostly through a MAO pathway except on a study on dopaminergic neurons describing as DAO GABA synthesis (PMID: 26430123) which was recently contradicted when they demonstrated that GABA released from dopaminergic neurons is mainly from previous reuptake (PMID: 35443174).

Manuscript was edited on lines 274-276

Meanwhile, SLC45A4 likely plays an important role in regulating GABA synthesis in all GAD-negative cells. The role of SLC45A4 in glial cells⁵⁵ and non-GABAergic neurons awaits further study.

Reviewer #2

Reviewer 2 commented that this paper is interesting, well-written and intuitive. Meanwhile, the

reviewer has some concerns. These comments are quite relevant and constructive. We want to take this opportunity to thank the review. And our responses are listed below.

Major critiques

Comment 1: *There is no direct in vitro data to show that SLC45A4 is a putrescine transporter. Many of the cellular tracing experiments in cells could be complex and indirect, as the authors have abundantly demonstrated. Furthermore, there is no guarantee that the inhibitors used in Fig 4 are specific, and potential non-specific effects of these inhibitors may affect data interpretation. The authors should attempt to perform direct reconstitution and transport experiments, as was previously described in Bartolke Biochem J 2014, to determine if reconstituted SLC45A4 is sufficient to confer putrescine transport in yeast or proteoliposomes. If direct reconstitution cannot be established, then a direct claim of a specific transporter activity cannot be made; rather, the claims should be modified eg “SLC45A4 is necessary for GABA synthesis in cells” etc.*

Response 1: We thank the reviewer for this remark on the direct evidence for putrescine transporting activity of SLC45A4. Indeed, we agree this is a key experiment. We used a wheat-germ cell-free translation system (CellFree Sciences, Ehime, JAPAN) to make reconstituted proteoliposomes containing SLC45A4. Our results show SLC45A4-containing proteoliposome has higher putrescine uptake activity compared to the control proteoliposomes, which contains a chimeric membrane protein eGFP-OMP25.

Manuscript was edited on lines 232-236:

Finally, we investigated what substrate is transported by SLC45A4. We used a wheat-germ cell-free translation system to make reconstituted proteoliposomes containing SLC45A4 or a chimeric membrane protein eGFP-OMP25 (Figure 5b). We performed a ^3H -putrescine uptake assay and our results show SLC45A4-containing proteoliposome has significantly higher putrescine uptake activity compared to the control proteoliposomes (Figure 5b).

Figure 5b. SLC45A4-containing proteoliposomes (SLC45A4) and Control (Ctr, OMP25-containing proteoliposomes) in presence of ^3H -putrescine.

Comment 2: *The data showing that SLC45A4 is mitochondrial is weak. First, the GFP tagged version in Fig 5 does not seem to co-localize with mitochondria. Second, the fractionation performed in (b) is complicated since lysosomes and other organelles can also co-fractionate with mitochondria. Third, SLC45A4 is not in mitocarta or other mitochondrial proteome*

compendiums. Fourth, SLC45A4 does not have an obvious mitochondrial targeting signal. My speculation would be that this transporter is actually in another compartment?

Response 2: We sincerely thank the reviewer for this critical remark on the subcellular localization of SLC45A4. We apologize again for the confusion we have made in the first submission. Our new results support the idea that SLC45A4 is a peroxisomal transporter rather than a mitochondrial transporter. The reason for the mistake we made was that the overexpression may have resulted in an excess amount of SLC45A4 protein and perhaps an abnormal localization. Therefore, we switched to a newly obtained antibody that can detect the endogenously SLC45A4. We believe this result better reflects the true physiological localization of SLC45A4.

Manuscript was edited on lines 221-230:

First, we tested the specificity of a commercial SLC45A4-targeting antibody by transiently overexpressing SLC45A4 with an HA-tag and observed perfect colocalization between green signals (anti-HA) and red signals (anti-SLC45A4) (Supp. Figure. 8). Therefore, we used this antibody to detect endogenous SLC45A4 transporter in A549 cells. SLC45A4 (green signal) is clearly detected around the cell nucleus, without overlapping actin signal (Figure. 5a., top panel), suggesting organellar localization rather than plasma membrane localization. Moreover, SLC45A4 signal strongly co-localized with ABCD3, which is a peroxisomal transporter of long-chain fatty acids and is frequently used as a peroxisomal marker (Figure. 5a, lower panel). In addition, we observed no overlap between SLC45A4 signals and Golgi marker, Giantin; endoplasmic reticulum marker, Calnexin or mitochondrial marker, Mitotracker (Supp. Figure. 9).

Figure 5. SLC45A4 encodes a peroxisomal putrescine transporter. a. Immunofluorescent detection of SLC45A4 subcellular localization. SLC45A4 endogenous, green; ABCD3, peroxisome marker, red.

Supplementary Figure 9. Immunofluorescent detection of SLC45A4 subcellular localization. SLC45A4 endogenous, green; organellar markers, Giantin (Golgi apparatus), Calnexin (Endoplasmic reticulum), and Mitotracker (Mitochondria) red.

Minor critiques

Comment 3: *As negative controls, it would also be informative to understand whether overexpression or knockdown of the other SLC45A1/2/3 members altered GABA levels in cells.*

Response 3: We have focused on SLC45A4 due to its strong correlation ($t=13.46$) with GABA. SLC45A1 ($t=0.30$), SLC45A2 ($t=-1.20$), and SLC45A3 ($t=4.52$) were not included in our study, due to their low correlation with cellular GABA levels. Our future endeavor will include the characterization of other SLC45 family members.

Comment 4: *The relevant cell type for GABA biosynthesis is not A549, H1299, or HEP62 cells, but rather GABAergic neurons. Does genetic manipulation of SLC45A4 in neurons also modulate GABA levels?*

Response 4: We thank the reviewer for this interesting comment. We agree that GABAergic neurons are the main producers of GABA. However, GABAergic neurons or microglia cells essentially produce their GABA from the GAD pathway which is not active in our cell models (Fig 2b). Based on our knowledge of the biochemical function of SLC45A4, we do not anticipate GABA synthesis through the GAD pathway is regulated by SLC45A4 in GABAergic neurons. However, in midbrain dopaminergic neurons, which were reported to produce GABA from putrescine through ALDH1A1 activity, SLC45A4 could be an important regulator. This is surely an interesting further direction in our lab.

Comment 5: *The data in Supplementary Fig 1 showing t-stat values between the 225 metabolites and 464 SLC genes is interesting. To help facilitate open data sharing, such data should be supplied as raw data in a searchable excel sheet.*

Response 5: This is an excellent suggestion. We have organized our data into a table, and it will be published as Supplementary Table 1 with this paper.

REVIEWER COMMENTS

Reviewer #2 (Remarks to the Author):

The authors have satisfactorily addressed my questions

Reviewer #3 (Remarks to the Author):

The manuscript by Colson C et al. reveals a novel function of the solute carrier protein SLC45A, which facilitates the transport of putrescine across peroxisome membrane, subsequently enhancing GABA *de novo* synthesis via the AOP pathway. However, the study's generalizability is limited due to the experimental data being generated solely from two lung cancer cell lines. Several critical questions remain unanswered regarding the impact of this enhanced GABA *de novo* synthesis by SLC45A on cellular physiology. Several concerns need to be addressed.

Major points:

Comment 1. *Lung cancer cell lines, including A549, which was used in this study appear to express GAD1 (doi: 10.1038/s41556-021-00820-9) and have been reported to synthesize and release GABA. How do the authors exclude the contribution from GAD1 to GABA production?*

Response 1: We thank the reviewer for this important comment. While we did not directly measure GAD1 expression levels in our cells, we used LC-MS to detect GABA derived from the GAD pathway. As shown in Figure 2b, when cells were supplemented with ¹³C₅-Glutamine, we observed strong ¹³C labeling in Glutamate (50-70%) but not in GABA (<1% in A549 and H1299 cells). Our results indicate that although glutamate decarboxylases (GAD1 and GAD2) play important roles in GABA production in GABAergic neurons, in A549 and H1299 cells they represent a functionally negligible route of GABA synthesis. More broadly speaking, Supplementary Figure 3b shows that GAD1 and GAD2 are only weakly positively correlated with the cellular GABA across 900 human cancer cell lines, while SLC45A4 has a strong positive correlation with cellular GABA (Figure 1a). We understand that this information could be critical and we emphasize this information in the manuscript, page 3 lines 129-132:

While supplemented ¹³C₅-glutamine produced a significant amount of ¹³C₅-glutamate (50-70%), it labeled less than 1% of GABA in A549 and H1299 cells (Figure. 2b), suggesting that glutamate decarboxylases (GADs) do not contribute significantly to GABA synthesis in these cells.

Comment 2. *Although LC-MS analysis can quantify GABA levels, it does not seem to allow direct comparison of GABA concentrations across different experimental runs (e.g. Fig 1b, right panel, HepG2 vs Fig 1c HepG2 Ctr). It would be useful to validate the findings using alternative methods, such as GABA ELISA or GABA-sensor or electrophysiology with GABAAR receptor-expressed cells, to determine the intracellular and extracellular GABA concentrations.*

Response 2: We agree with the reviewer that LC-MS is generally not used for an absolute quantification of GABA levels between different experiments. Instead, LC-MS typically reports "Ion Count" in an arbitrary unit, corresponding to the compound's level of ionization within the experiment. Several parameters could influence the Ion Count, such as the number of cells

extracted. This is the reason Figure 1b and Figure 1c cannot be directly compared since they were from different batches of cells with different numbers. However, LC-MS measurements of GABA serve an important role in our study. It allows us to use different tracers to label the synthetic precursors of GABA to quantitatively assess GABA synthesis and uptake. Neither ELISA nor GABA-sensor would allow such analysis.

We do agree with the Reviewer that an absolute quantitation of GABA could establish a benchmark and facilitate future inter-study comparisons. To achieve this goal, we spiked $^{13}\text{C}_4$ -GABA into the cellular and media extracts at the concentration of $1\ \mu\text{M}$. By measuring the ratio of endogenous ^{12}C -GABA and exogenous $^{13}\text{C}_4$ -GABA, we calculated the intracellular GABA production to be $1.67\ \text{nmols/millions}$ (10^6) cells and $0.85\ \text{nmols}/10^6$ cells in A549 and H1299 cells respectively (Supp. Figure. 4a, and below).

Supplementary Figure 4. GABA quantitation in lysates and media from A549, H1299 and HepG2 cells. Endogenous GABA concentration (^{12}C -GABA) in lysates (a.) and media (b.) from A549 and H1299 cells measured by LC-MS and $^{13}\text{C}_4$ -GABA as internal standard.

To further clarify the GABA measurements and the absolute quantitation, we edited the manuscript accordingly, page 3 lines 90-92:

Indeed, the absolute quantitation using ^{13}C internal standard showed that the GABA concentration in A549 cells is $1.67\ \text{nmols/millions}$ of cells (Supp. Figure 4a.).

Comment 3. *The authors propose the conversion of external putrescine to GABA, which is then transported into the cells. However, several proteins, in addition to canonical GABA transporters, could mediate GABA transport across the membrane, such as TauT, BGT, BEST, VRAC. The authors should investigate this possibility. Notably, a GAT2/BGT1 inhibitor is also available (<https://doi.org/10.1016/j.neuint.2005.12.031>).*

Response 3: We thank the reviewer for the suggestion of investigating non-canonical GABA transport in SLC45A4-mediated GABA production. Among the suggested transporters, some have been described as direct GABA transporter (TauT/SLC6A6, BEST/GAT1/SLC6A1 and BGT1/SLC6A12) or GABA transport facilitators (VRAC/LRRC8A). In the CCLE database, we see that only TauT/SLC6A6 and VRAC/LRRC8A are more expressed than SLC45A4 in our cell models; however, none of them are significantly correlated with GABA levels (See below Figure R1).

Although these transporters can participate in alternative GABA transport, our results show that they are not the major determinants of cellular GABA levels. While our results in **Figure 1e** demonstrate cells are capable of taking extracellular GABA, which was $^{13}\text{C}_4$ -labeled, SLC45A4 expression does not correlate with GABA uptake. Meanwhile, **Figure 1f** clearly shows that it

was the endogenous (^{12}C -GABA) that was significantly decreased by SLC45A4-KD. Since our main focus is to explain the role of SLC45A4 in GABA metabolism, we concluded that SLC45A4 promotes GABA *de novo* synthesis rather than acting as a GABA transporter. Whether other GABA transporters have synergistic effects with SLC45A4 is an interesting question for sure, but is beyond the scope of the current manuscript.

Figure R1. Non-canonical GABA transporters. **a.** CCLC database RPKM for non-canonical GABA transporters and SLC45A4 (far right). **b.** CCLC association analysis between GABA levels and transporters expression level.

Comment 4. *The study was conducted in only two lung cancer cell lines. It would be valuable to assess GABA levels in other cancer cell lines or primary cells with high SLC45A expression to further validate the findings.*

Response 4: We thank the reviewer for this remark. A549 and H1299 cell lines were selected for their relatively high expression levels of *SLC45A4*. However, our initial result which suggested to us that *SLC45A4* correlates with cellular GABA levels was derived from the CCLC database, which contains 928 human cancer cell lines of various lineages. In addition, our transcriptomic-metabolomic analysis from CCLC also identified *ODC1* as the second highest gene correlated with GABA levels among <56,000 transcripts (Supp Figure 3c), validating that GABA synthesis mediated by *SLC45A4* is generally processed through AOP pathway such as described in this study. We agree with the reviewer that investigating this mechanism in primary cells will bring important information regarding this pathway in healthy conditions and we added this discussed information in the discussion page 7, lines 273-276:

Our transcriptomic-metabolomic association analysis established that cellular levels of creatine, carnitine, and taurine are mostly determined by the expression of their corresponding transporters. This result indicates that in many cases, *in vitro* cultured cells don't perform *de novo* synthesis of these metabolites but instead take them from the media, presumably from the

serum component. In light of this, it is conceivable that dialyzed FBS, compared to regular FBS, may affect cell metabolism due to its lack of key metabolic factors. Meanwhile, one potential caveat of our analysis is that it is based on the metabolomic profiles of cancer cells cultured under common media such as DMEM or RPMI 1640. It would be important to validate such correlations using either primary cells and/or culture media resembling tumor interstitial fluid.

Comment 5. *In Figure 2c, is it correct that ^{13}C -arginine labeled approximately 60% GABA with ^{13}C in HepG2 cells?*

Response 5: We thank the reviewer for this observation and we confirm that in Figure 2c, 60% of GABA is ^{13}C -labeled in HepG2 cells. Data are normalized to compare cell lines but it represents extremely low ion counts from HepG2 cells compared to A549 and H1299 cells.

Comment 6. *Several protein bands for SLC45A4 are shown in Figure 3b. The specificity of this antibody should be further validated using other cell or tissue lysate with varying SLC45A4 expression levels..*

Response 6: We want to thank the reviewer for this comment and agree that SLC45A4 antibody shows unspecific pale bands. This could be explained by its polyclonal nature. In addition, SLC45A4 transporter is a transmembrane protein which increase its interaction with partners compared to cytosolic proteins and so, increase the possibility for unspecific signals. However, the major signal is displayed around 100 kDa which tends to the theoretical size of SLC45A4 (around 84 kDa); this signal is blunted in SLC45A4-KO cells (KO1&KO2) and highly increased in cells overexpressing SLC45A4 (OE) (Figure 3b. and below). These results strongly confirm that our antibody specifically detects SLC45A4 protein.

b.

Figure 3. GABA production from ornithine and putrescine is differentially regulated by SLC45A4. **b.** SLC45A4 protein expression in A549 and H1299 WT, overexpressing SLC45A4 (OE) and KOs cells.

Comment 7. *On page 5, line 192, it is stated: "All the DAO inhibitors showed inhibition of GABA production from 2H-ornithine, but neither of the MAO inhibitors did (Figure 4e)". However, did clorgyline significantly reduce 2H-GABA, as shown in figure 4e right panel?*

Response 7: We thank the reviewer for this remark. We agree that clorgyline induces a small but significant decrease of ^2H -GABA. While clorgyline is described as a MAO inhibitor, we cannot completely rule out a small inhibitory effect on DAO. In addition, selegiline had no effect on ^2H -GABA levels while described DAO inhibitors pentamidine and aminoguanidine completely blunted 2H-GABA production. In order to avoid overstating, we edited the manuscript accordingly; page 5 lines 194-196:

While clorgyline induced a small decrease in ^2H -GABA, selegiline showed no effect on GABA production when both the DAO inhibitors more profoundly blunted GABA production from $^2\text{H}_6$ -ornithine (Figure. 4e).

Comment 8. In figure 5, the staining of SLC45A4 appears heterogenous, with not all cells expressing SLC45A (especially in the right panel). Immuno-EM would help confirm the

Response 8: We thank the reviewer for this comment and we agree that SLC45A4 expression could appear heterogenous between cells in the bottom panel of Figure 5a. (previously right panel). While it could be directly SLC45A4 expression involved, these images are zoomed-in in order to better display the SLC45A4 subcellular localization and be more representative; thus, they don't show all the cells. Supplementary Figure 9 also display different levels of expression for SLC45A4.

In addition to the immunofluorescence detection of peroxisomal SLC45A4, we have a successful isolation of peroxisomes from A549 cells using 2 different methods: 1) Immunoprecipitation with magnetic HA-beads (Thermo, 88837) of A549 cells expressing the peroxisome membrane bound protein PEX26 tagged with eGFP and 3xHA (PEX26-eGFP-3xHA) and now presented in the main figures of the manuscript as **Figure 5b to 5d**. (See below), and 2) Differential centrifugations and iodixanol gradient (Sigma PEROX1) of A549 cells and now presented in **Supplementary Figure 11b** and below. In both isolations, the results show that SLC45A4 protein is enriched in the subcellular fraction containing ABCD3 and catalase proteins which confirm SLC45A4 localization in the peroxisome.

[editorial note: panel d partially redacted due to third-party material]

Figure 5. SLC45A4 encodes a peroxisomal putrescine transporter. b. Scheme of the construct containing a truncated membrane bound peroxisome protein PEX26 (237-305) tagged with e-GFP and 3-HA. **c.** A549 cells stably expressing the PEX26-eGFP-3xHA (top panel) co-localizing with peroxisome marker ABCD3 (bottom panel). **d.** Simplified workflow for HA immunoprecipitation (HA-IP) from A549 cells stably expressing PEX26-eGFP-3xHA and SDS-PAGE of whole cell, total lysate and HA-IP fraction: SLC45A4 was detected with markers of peroxisome, ABCD3 and catalase.

Supplementary Figure 11. SLC45A4 protein is enriched with peroxisomal proteins. b. SDS-PAGE of A549 cells sub-fractionated using differential centrifugation and density gradient.

We have edited the manuscript according to these new results page 6 and lines 235-247:

To further validate the subcellular localization of SLC45A4, we isolated peroxisomes in A549 cells by using two different methods, either with an affinity tag (Figure. 5b to 5d) or with the more traditional density gradient centrifugations (Supp. Figure. 10a and 10b). We first used A549 cells stably expressing a truncated peroxisome membrane protein PEX26 tagged with e-GFP and 3xHA⁵² (Figure 5b) and we confirmed that the chimeric protein co-localizes with a specific peroxisome membrane-bound protein ABCD3 (Figure 5c). A549 cells PEX26-eGFP-3xHA were submitted to hypotonic lysis and immunoprecipitation with magnetic beads tagged with anti-HA antibody, following the protocol developed in Sabatini's lab⁵² (Figure 5d, top schematic) and the results showed a strong enrichment of SLC45A4 protein in the HA-IP fraction, associated with an enrichment of peroxisome-specific proteins ABCD3 and catalase. These results were corroborated with gradient density centrifugation protocol (Supp. Figure. 11a) which showed similar SLC45A4 enrichment in the fractions containing peroxisome specific proteins, ABCD3 and catalase (Supp. Figure. 11b; lanes L1 and L2) compared to lysates pre-gradient separation (Supp. Figure. 11b; lanes WC, P1 and Sup1).

Comment 9. *Both cell lines used in this study express functional GABA receptors, and exogenous GABA has been shown to affect their proliferation (doi: 10.1186/1479-5876-11-102, <https://doi.org/10.1038/srep32756>), it would be interesting to investigate whether de novo synthesized GABA by SLC45A4 acts its own receptor in an autocrine manner.*

Response 9: We want to thank the reviewer for this critical comment. Our study already show that cells are able to produce and release GABA since we detect labeled GABA in the extracellular media. Therefore, it will be really interesting to investigate the destiny and role of this extracellular GABA, as an autocrine/paracrine or endocrine factor. Our future work will investigate these possibilities.

Minor points:

Comment 1. *The ^{13}C -GABA concentration in Figure 1d is reported as 20 μM , which differs from the legend (5 μM). Please clarify.*

Response 1: We are thankful to the reviewer for noticing this important information and we clarified the figure and its legend as ^{13}C -GABA was used at 5 μM .

Comment 2. *Page 3, line 116, ^{13}C -GABA is significantly reduced in A549KD cells (Fig 1e right panel).*

Response 2: We thank the reviewer for spotting this and we changed the text page 3, lines 118-119 according to the figure.

The $^{13}\text{C}_4$ -GABA, which indicates GABA uptake, was significantly but slightly changed in A549 KD cells while not at all in H1299 KD cells.

Comment 3. *For improved statistical reliability, it would be preferable to include at least 5 independent replicates rather than 3.*

Response 3: We thank the reviewer for his comment and we believe 3 independent replicates are sufficient to present clear results as we observed the same results in several independent experiments.

Comment 4. *Figure 4g, are the groups with 100 nM aminoguanidine statistically significant? Please clarify*

Response 4: We thank the reviewer for his comment and we agree that this part needed clear statement. While the changes appear small, 100 nM of aminoguanidine significantly reduced ^{13}C -GABA fraction while significantly increased ^2H -GABA fraction in A549 and in H1299 as shown in Supplementary Figure 8a right panel. The actual ratio between the two fractions is shown in Figure 4f and Supplementary Figure 8a, left panel. We have added the information on significance in the manuscript, lines 205-207:

As previously shown in Figure 4e., aminoguanidine decreases total cellular GABA but $^2\text{H}/^{13}\text{C}$ ratio increases by 2 folds (Figure. 4f, Supp. Figure. 8a), indicating that GABA derived from extracellular putrescine (^{13}C fraction) is significantly reduced (10% reduction, Figure. 4g) while GABA derived from intracellular putrescine (^2H fraction) is significantly augmented (2 folds, Figure. 4g and Supp. Figure. 8a).

Comment 5. *Page 6, line 225, left panel; line 228, right panel.*

Response 5: We sincerely thank the reviewer for this important observation and as we edited the Figure 5 it was referring to, it is now appropriate to say “top panel” and “bottom panel”.